# Merging conformational landscapes in a single consensus space with FlexConsensus algorithm

**David Herreros** [1] ✉, **Carlos Perez Mata** [2], **Carlos Oscar Sanchez Sorzano** [1,3] **& Jose Maria Carazo** [1,3]

Structural heterogeneity analysis in cryogenic electron microscopy is experiencing a breakthrough in estimating more accurate, richer and interpretable conformational landscapes derived from experimental data. The emergence of new methods designed to tackle the heterogeneity challenge reflects this new paradigm, enabling users to gain a better understanding of protein dynamics. However, the question of how intrinsically different heterogeneity algorithms compare remains unsolved, which is crucial for determining the reliability, stability and correctness of the estimated conformational landscapes. Here, to overcome the previous challenge, we introduce FlexConsenus: a multi-autoencoder neural network able to learn the commonalities and differences among several conformational landscapes, enabling them to be placed in a shared consensus space with enhanced reliability. The consensus space enables the measurement of reproducibility in heterogeneity estimations, allowing users to either focus their analysis on particles with a stable estimation of their structural variability or concentrate on specific particle subsets detected by only certain methods.

Cryogenic electron microscopy (cryo-EM) is experiencing a paradigm shift in exploring conformational variability using experimental data. Unlike classical three-dimensional (3D) classification algorithms[1] that are confined to a set of reduced and stable states, new heterogeneity algorithms focus on estimating richer and more complete conformational landscapes with the possibility of retrieving any conformation from the approximated continuum.

Heterogeneity algorithms can be classified based on the information estimated from cryo-EM data. Heterogeneous reconstruction methods[2–5] rely on the direct estimate of electron density maps from a continuous function that maps every point in the conformational landscape to a 3D volume. On the other hand, deformation field-based methods directly estimate the motions responsible for driving a reference state to a new conformational state represented in a particle dataset[6–8].

The variety of methods, approaches and implementations allows one to explore the structural variability of any given dataset through a systematic approach. However, the large pool of existing algorithms also presents the challenge of comparing different results. Currently, there is a lack of validation tools in the cryo-EM heterogeneity field. Indeed, the most widely used validation approach is to reconstruct states from a small subset of images surrounding a given conformation in the landscape and to compare them with the conformations estimated by a given heterogeneity analysis method. However, this approximation is limited by the minimum number of particles needed to reconstruct a volume with enough resolution to detect a given conformational state. If the number of images required is too large, contamination from other structural changes in the reconstruction will prevent a complete validation of the conformational space. Furthermore, the accuracy of current heterogeneity analysis methods

[1]Centro Nacional de Biotecnologia-CSIC, Madrid, Spain. [2]PKF Attest innCome, Madrid, Spain. [3]These authors jointly supervised this work: Carlos Oscar Sanchez Sorzano, Jose Maria Carazo. ✉e-mail: dherreros@cnb.csic.es

remains a subject of ongoing investigation. Therefore, it is crucial to extract consensus solutions from different approaches.

To overcome the challenge of comparing conformational landscapes obtained from different methods and/or runs, we propose in this work a new deep learning algorithm: FlexConsensus. FlexConsensus introduces a multi-autoencoder architecture designed to merge different conformational landscapes into a common latent space, which is subsequently decoded back into the original representations input to the network. In this way, it is possible not only to analyze the common conformational space but also to determine a consensus metric that measures the stability of every estimation in the original conformational landscapes to filter out only those regions with higher confidence. Lastly, the method also allows for converting among different conformational landscape representations, simplifying the comparison of techniques.

FlexConsensus proposes a framework that is not aimed at estimating states from cryo-EM data but, rather, focuses on mapping already estimated landscapes into a common consensus space while preserving the organization of the original landscapes and enhancing their interpretability. Therefore, the application of FlexConsensus does not involve computing or modifying a given conformational space with a priori information, as suggested by other methods, such as the incorporation of structural priors[9], the disentanglement of the space into meaningful components[4] or methods employing statistical approaches to analyze the landscape[10], among others. Instead, analyzing the consensus spaces facilitates the identification of similarities and differences among various methods, assisting in the validation of estimations and providing tools to streamline the heterogeneity workflow based on the derived reliability scores.

## Results

The following sections present some use cases to evaluate and discuss the precision and performance of FlexConsensus under different conditions. The analysis starts by evaluating the capabilities of FlexConsensus with two datasets from CryoBench[11], followed by two experimental datasets to assess its behavior in more realistic and challenging scenarios.

In the Supplementary Information, we include three additional analyses using simulated data, which provide a deeper understanding of the network's training workflow and the characteristics of the consensus space.

We note that FlexConsensus learns by default a consensus space with the number of dimensions set to the minimum dimension among the input spaces. However, the latent space dimensionality is exposed as a customizable parameter in the corresponding Scipion[12] protocol form. It is important to note that setting a consensus space dimension smaller than the intrinsic dimensionality of the input latent dimensions means that some information from the inputs will potentially be ignored when mapping to the consensus space.

All the consensus landscapes presented in the following sections were defined to have three dimensions, thereby avoiding the need for any further dimensionality reduction step.

The workflow for training the network is described in detail in the Methods section and Fig. 6.

All the data were analyzed with the Scipion 3.8.0 software package. Inside Scipion, CryoDRGN version 3.4.0 was also used to estimate the conformational landscapes analyzed in this paper as well as Flexutils plugin version 3.3.0.

CryoSPARC 4.5.1, Relion 4.0 and Xmipp 3.24.12.0 packages were also used to preprocess the data.

### Consensus results on CryoBench datasets

Before moving to testing FlexConsensus on experimental data, it is interesting to check its performance on controlled and challenging 'standardized datasets' capturing different ranges of motion and variability. Therefore, we tested FlexConsensus on two different datasets included in CryoBench[11].

The first dataset to be explored is the IgG-RL one, representing a set of random conformations for a disordered peptide linker connecting the Fab to the rest of the IgG complex. This dataset corresponds to a challenging disordered motion, which is very difficult to follow by all proposed heterogeneity analysis methods thus far, as indicated in the original CryoBench paper. Indeed, the conformational landscapes obtained by the different methods have notable differences. It is an excellent example for integrative tools such as FlexConsensus, where we can explore the similarities and differences among different methods.

The proposed experiment starts by estimating the conformational spaces using two different methods: HetSIREN and CryoDRGN. To simplify the FlexConsensus results analysis, HetSIREN was forced to estimate only a conformational latent space without refining the original angles, so its way of working is similar to CryoDRGN. The landscapes predicted by each method were then fed into FlexConsensus to estimate the consensus landscape. The results from this analysis are summarized in Fig. 1. The consensus space obtained from the two original estimations shows a clear difference in the distribution of states found by the two methods. This can be further quantified by analyzing the consensus errors measured in the consensus space.

Thus, consensus metrics from the previous analysis were used to filter the landscape toward a stabilized representation with a more reliable state distribution. Based on the consensus, it is possible to define a statistical framework to determine a significant threshold based on the similarity of the distribution of states represented by the different consensus landscapes. To that end, it is possible to work under the assumption that the distributions of states of different methods in the consensus space should be the same, allowing us to derive the threshold that fulfills the previous assumption.

Following the previous reasoning, we proposed an approach based on FlexConsensus to derive the previous threshold based on a random permutation test of the distances between identical samples in the consensus space. It allows us to obtain a $P$ value representing the probability of measuring the distances observed in the consensus space against random labeling. The Methods section describes this filtering process and the threshold computation from the previous $P$ value in more detail.

The application of the previous test to the IgG-RL consensus space is summarized in Fig. 1. At the beginning of the figure, we show the unfiltered consensus spaces, followed by the plot of the $P$ values that the previously described test yielded. Additionally, we show below the previous plot the permutation distribution of distances for the selected threshold, showing that most of the measurements computed from the random labeling fall below it. After applying the threshold highlighted in these plots, it is possible to get a representation of the consensus space, including only those images estimated to have a similar conformational state according to the two input methods.

For this experiment, the filtering process suggests that the distribution of states in the original spaces was, in general, not similar, which aligns with the findings found in the original CryoBench paper[11]. This is shown when checking the threshold found with this test, as presented in the '$P$ value analysis' plot, which translates into a large fraction of discarded images (only approximately 10,000 images are kept after applying the threshold). Additionally, the filtered space isolates more independent antibody states than the original consensus, helping identify the different motions simulated in the dataset. In Supplementary Video 1, we inspect the filtered consensus space and the associated conformational states.

To better quantify the relationship between the antibody's position and the location in the filtered consensus space where potentially different conformational states are more easily detected, we performed an additional analysis to determine if the previous two variables are

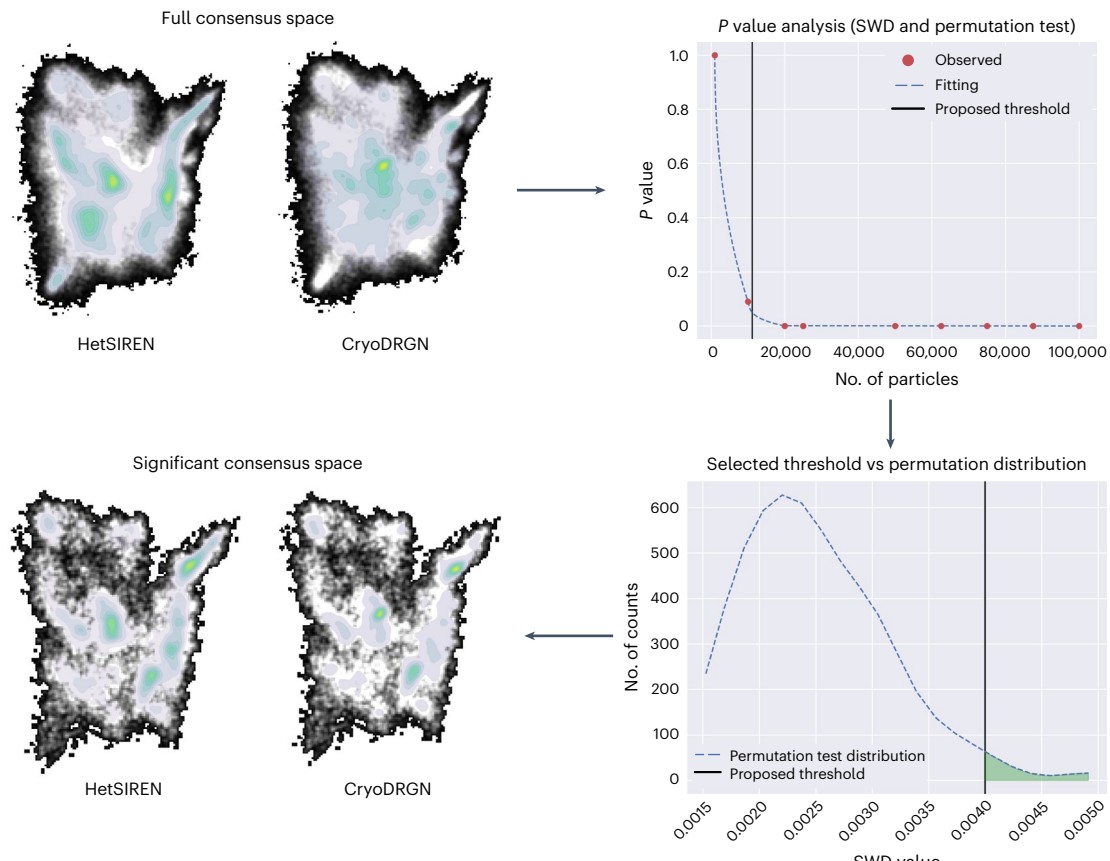

**Fig. 1 | FlexConsensus analysis resulting from comparing the estimations of HetSIREN and CryoDRGN for the IgG-RL dataset included in CryoBench[11].** The image shows both the original consensus space derived from the two methods and the consensus space obtained by keeping only those particles in the consensus space that are significantly similar. The first two landscapes displayed in the image correspond to the unfiltered consensus spaces obtained with FlexConsensus. Following these spaces, we include the plot with the different $P$ values obtained from the permutation test when filtering an increasingly larger number of samples, highlighting the threshold proposed by the test (which corresponds to a $P$ value of 0.05 from the one-sided test). Below the previous plot, the permutation distribution of distances for the selected threshold is included, showing that most of the measurements computed from the random labeling fall below it. Lastly, the figure shows the consensus spaces obtained after applying the threshold proposed by the test, showing a more similar distribution of states. SWD, sliced Wasserstein distance.

correlated. To that end, the filtered consensus space was split into 10 uniformly distributed clusters with $k$-means. The representatives of each cluster were then converted to HetSIREN latent space vectors using the decoding capabilities of the FlexConsensus network, allowing us to recover the conformational states associated with the cluster representatives. Each volume was segmented using *Segger*[13] to isolate the heavy and light chains of the protein. Because of this segmentation, it is possible to measure the center of mass of the fixed heavy chain and the center of mass of the light chain at the different conformations. To characterize the motion of the antibody, we measured the distance from the center of mass of the heavy chain to the center of mass of the light chain as well as the angle defined by the previous two centers of mass and the anchor point between the heavy and the light chains.

The next step was to characterize the presence of a correlation between the two measured variables (distance between centers of mass and angle) and the location in the filtered FlexConsensus space. To that end, we carried out a Mantel test, working under the hypothesis that the distance matrices defined by the FlexConsensus latent vectors and the two variables measured in the volumes are correlated. A summary of this experiment is provided in Extended Data Fig. 1a. As seen from the figure, the Mantel test returns a $P$ value of 0.002, confirming the hypothesis that there is a correlation between the motions of the antibody at the level of maps and the location in the consensus space.

The second dataset explored was the MDSPike dataset included in CryoBench. This dataset was simulated from a long-timescale

molecular dynamics simulation, yielding a free-energy landscape that was sampled to generate approximately 50,000 structures. These structures were then converted to electron density maps and projected to create the images in the dataset. The ground truth locations of these structures in the energy landscape are also provided.

Similar to the case before, HetSIREN (with no pose refinement) and CryoDRGN were executed to estimate the conformational space from this dataset, which was then analyzed with FlexConsensus. The resulting consensus spaces are shown in Fig. 2. As can be seen from the figure, the distribution of states estimated by the two methods was more similar in this case, which also translates into a filtered landscape losing a lower fraction of particles. In Supplementary Video 2, we provide an inspection of the filtered consensus space and the associated conformational states, which shows the transition from the one-up receptor-binding domain (RBD) state to the three-down state simulated in this dataset.

Apart from comparing estimations from different methods, the MDSPike dataset opens an interesting possibility for evaluating how methods compare to molecular-dynamics-determined energy landscapes. Therefore, we performed additional analysis to map experimental and simulated landscapes onto a common space, focusing on the possibility of correlating collective variables characteristic of a given motion with the consensus space. To that end, we trained a new FlexConsensus network with the ground truth latent coordinates obtained from the simulation and the experimental landscapes

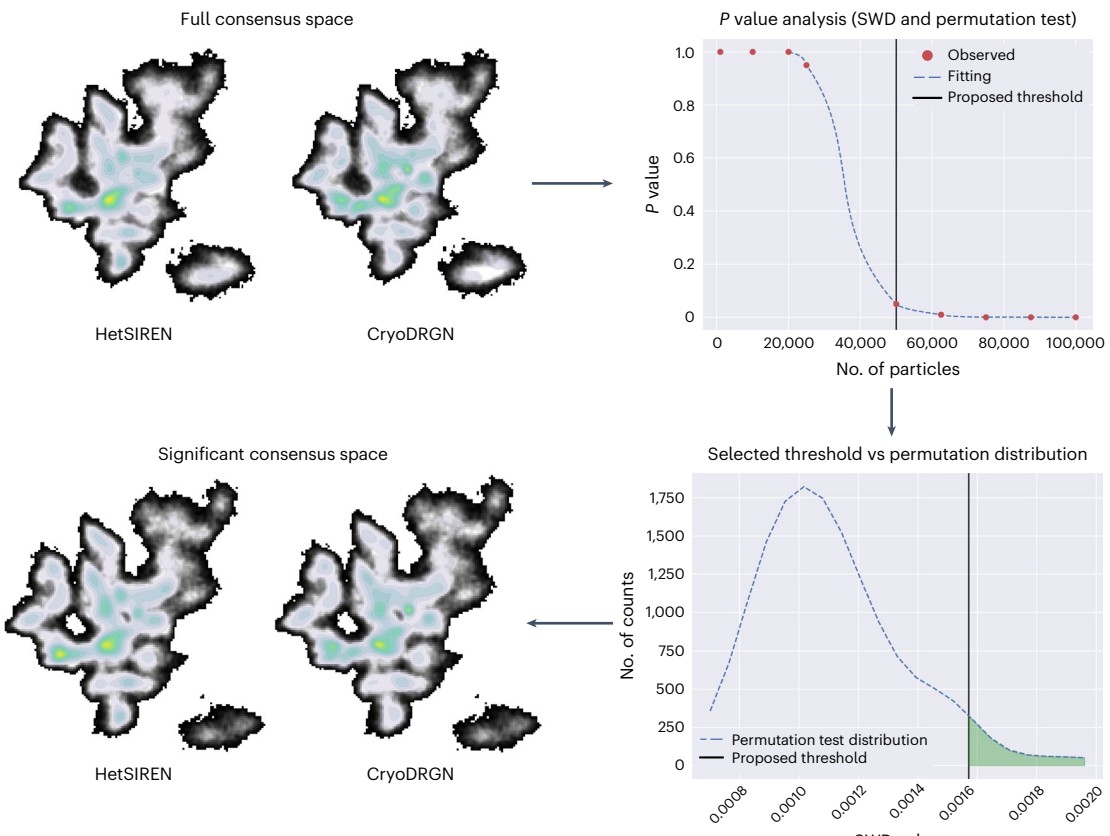

**Fig. 2 | FlexConsensus analysis resulting from comparing the estimations of HetSIREN and CryoDRGN for the MDSPike dataset included in CryoBench[11].** The image shows both the original consensus space derived from the two methods and the consensus space obtained by keeping only those particles in the consensus space that are significantly similar. The first two landscapes displayed in the image correspond to the unfiltered consensus spaces obtained with FlexConsensus. Following these spaces, we include the plot with the different $P$ values obtained from the permutation test when filtering an increasingly larger number of samples, highlighting the threshold proposed by the test (which corresponds to a $P$ value of 0.05 from the one-sided test). Below the previous plot, the permutation distribution of distances for the selected threshold is included, showing that most of the measurements computed from the random labeling fall below it. Lastly, the figure shows the consensus spaces obtained after applying the threshold proposed by the test, showing a more similar distribution of states. SWD, sliced Wasserstein distance.

obtained with HetSIREN and CryoDRGN. Because the ground truth space was defined to have two dimensions, the FlexConsensus latent space was set to have this number of dimensions.

Once the network was trained, we took different latent vectors in the ground truth space along the two principal axes of the energy landscape, allowing us to define our collective variables. The extracted data were then input to FlexConsensus to map them in the consensus space, allowing us to evaluate their correlation with the consensus axes. The results of this experiment are summarized in Extended Data Fig. 1b. As seen from the figure, the collective variables extracted from the principal axes of the ground truth space have a strong correspondence with the main axis of the consensus space, implying that the consensus space has been able to capture the relevant structural information from the simulated motions.

### Consensus results on the EMPIAR 10028 dataset

The next step in evaluating the capabilities of FlexConsensus is applying the method to a more realistic scenario. Therefore, we proposed the evaluation of FlexConsensus with the EMPIAR 10028 dataset[14], a well-known and well-studied dataset showing different conformational states of the *Plasmodium falciparum* 80S ribosome bound to emetine. This dataset has been widely applied as a test case for the most recently developed heterogeneity methods, yielding a conformational landscape with well-defined features. In addition, the experimental images in the dataset mainly capture continuous conformational changes, although there is also a compositional variability component. Because

of all these characteristics, the EMPIAR 10028 dataset supposes a realistic yet controlled scenario to evaluate the consensus when considering conformational landscapes estimated by different methods.

The dataset was first preprocessed with CryoSPARC[15] inside Scipion[12] to yield a set of appropriately characterized experimental images to be analyzed by heterogeneity algorithms. This involves estimating contrast transfer function (CTF) information and particle alignments, which were subjected to a consensus analysis to improve their stability[16].

A set of approximately 50,000 particles was obtained from the preprocessing step and further analyzed for conformational variability. The study was carried out with two different methods: HetSIREN[5] and Zernike3D[6]. These two methods follow very different approaches to estimating conformational variability, HetSIREN being a heterogeneous reconstruction/refinement algorithm and Zernike3D being a deformation field-based method. Therefore, HetSIREN can extract continuous and compositional variability, whereas Zernike3D focuses on extracting continuous motions. Moreover, each method defines a conformational landscape with different dimensions, similar to the synthetic datasets analyzed in the previous sections.

In total, three independent conformational landscapes were estimated, corresponding to the execution of HetSIREN in reconstruction mode, the execution of HetSIREN in refinement mode and the execution of Zernike3D. The main difference between the reconstruction modes of HetSIREN is the algorithm's initialization: in reconstruction mode, the initial volume required by HetSIREN is initialized to have

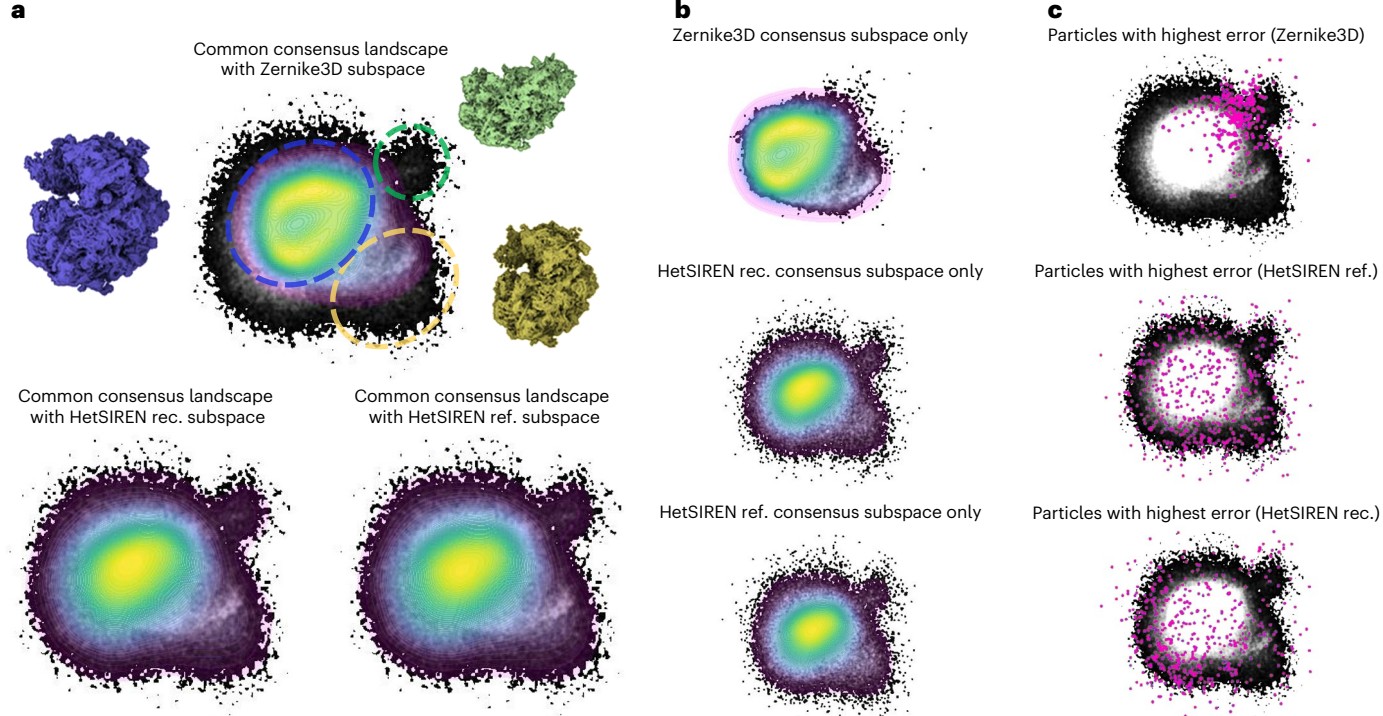

**a** Common consensus landscape with Zernike3D subspace

Common consensus landscape with HetSIREN rec. subspace

Common consensus landscape with HetSIREN ref. subspace

**b** Zernike3D consensus subspace only

HetSIREN rec. consensus subspace only

HetSIREN ref. consensus subspace only

**c** Particles with highest error (Zernike3D)

Particles with highest error (HetSIREN ref.)

Particles with highest error (HetSIREN rec.)

**Fig. 3 | Consensus landscapes obtained by FlexConsensus for the EMPIAR 10028 dataset[14].** Three conformational spaces were input to the FlexConsensus analysis, obtained from three independent runs of the following methods: HetSIREN in reconstruction/refinement mode[5] and Zernike3D[6]. Panel **a** compares the common consensus landscape (in grays) against each consensus subspace generated from each input (colored). The comparisons show a region identified by both HetSIREN runs that is not present in Zernike3D, corresponding to compositional changes of the ribosome. It should be noted that, due to the approximation of each method, HetSIREN can identify compositional variability, whereas Zernike3D focuses on continuous conformational changes.

Panel **b** displays the three subspaces presented in **a** individually, without being superimposed on the common consensus space. Panel **c** highlights the regions assigned a higher consensus error for each method. For Zernike3D, the highlighted regions correspond to the region where the biggest compositional variability change has been detected by HetSIREN, which is expected, as Zernike3D cannot properly identify compositional variability. In the case of HetSIREN, these regions are located on the border regions of the 3D consensus landscape that are farther from the main cloud. rec., reconstruction; ref. refinement.

only zero values. By contrast, HetSIREN in refinement mode receives as the initial volume a map calculated from all initial images. These landscapes were then fed to FlexConsensus to generate the consensus landscape and the error metrics needed to determine the reliability of the three estimations.

It is important to mention that FlexConsensus also works with estimations obtained from different executions of the same method under the same conditions, allowing the study of the consistency and reliability of a single algorithm's estimations. Although we could have executed HetSIREN in this way, we considered comparing this method's two modes of operation to determine if they impact the estimated conformational landscapes.

Before analyzing the results obtained for this dataset, we include some terminology that will be used from now on when comparing different results. We will refer to the space computed from the density distribution of the points in the consensus space associated with method X as 'X subspace'. These subspaces will help identify the location of the points coming from input space X in the common consensus space, allowing us to quickly compare the differences among the different methods in the consensus. The consensus space, composed of the addition of all the involved subspaces, will be referred to as 'common consensus space'.

The results obtained from the analysis of the common consensus space learned by FlexConsenus are summarized in Fig. 3. Figure 3a presents the common space with the previously described subspace associated with each input space (note that this two-dimensional image of consensus space corresponds to a selected view of the computed

3D space; full access to 3D space can be done through Flexutils, as described later on in the main text). At the top of Fig. 3a, the 'Zernike3D subspace' is presented, showing only those points coming from the Zernike 3D landscape (that is, the mapping onto the common space of the point cloud corresponding to the Zernike3D analysis) colored on top of the common consensus space (shown in grays). At the bottom, a similar representation is followed by the points coming from the two HetSIREN landscapes. The general disposition of the three subspaces is visually similar, indicating that the strategy of mapping all input results to the same common space is adequate. In detail, a difference between Zernike3D and HetSIREN is noticeable in the common landscape at the top rightmost corner, which is further highlighted by a circle drawn with a broken green line. The Zernike3D subspace in this area is unstructured (points are sparse), unlike both HetSIREN landscapes, which show highly ordered subspaces. Interestingly, the maps' analysis from these points indicates that they correspond to specimens presenting a substantial compositional variation (they lack the 40S subunit, showing only the 70S one), which Zernike3D, by design, could not capture. The point clouds within the blue and yellow broken line circles correspond to particles presenting internal motions, which all three methods have captured well. To simplify the visualization of each subspace in Fig. 3a, we include in Fig. 3b the subspaces of each method by themselves. In addition to the results displayed in Fig. 3a, we include in Fig. 3c a highlight of the regions estimated to be less reliable for every method according to FlexConsensus. In Zernike3D, the higher errors are associated with the compositional components, which agrees with the results previously discussed. In the case of HetSIREN, these regions

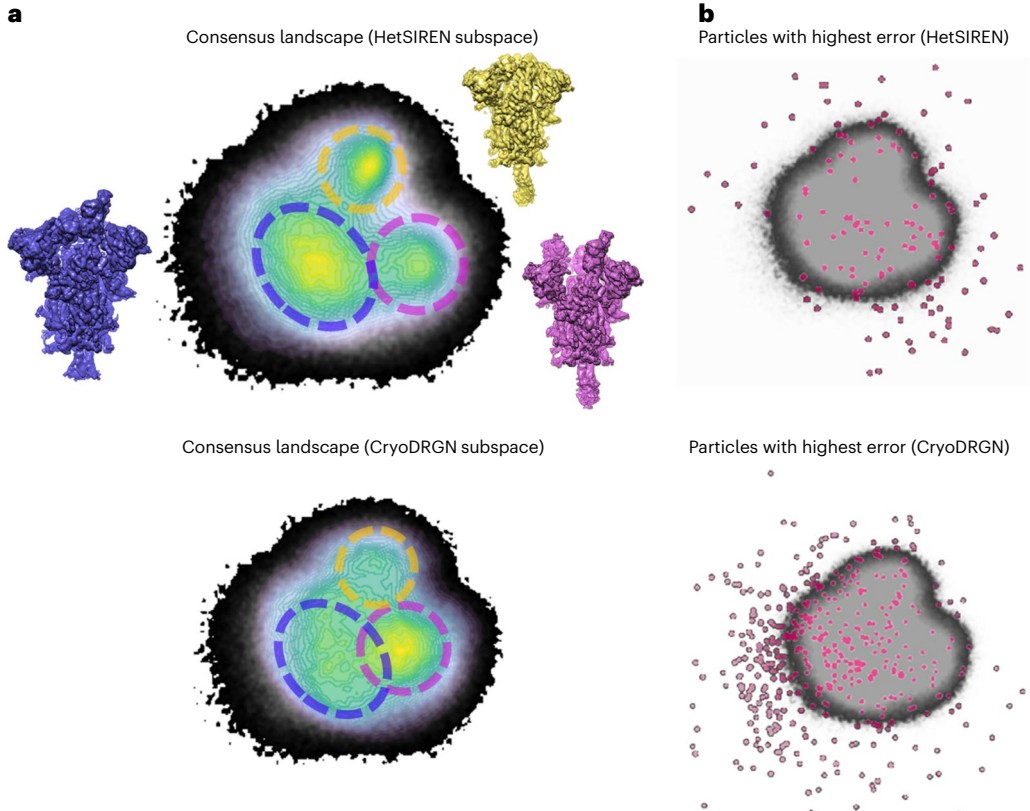

**Fig. 4 | Consensus landscapes obtained by FlexConsensus for the SARS-CoV-2 D614G spike variant.** Two conformational spaces were input to the FlexConsensus analysis, obtained from two independent runs of the following methods: HetSIREN in reconstruction mode[5] and CryoDRGN[2]. Panel **a** compares the common consensus landscape (in grays) against each consensus subspace generated from each input method (shown in colors). The consensus landscapes show three main regions corresponding to three different conformational states of the spike RBD: three-down, one-up and two-up. Based on the consensus, it is possible to see that both methods correctly identify the three structural states. However, HetSIREN is more evenly distributed than CryoDRGN, which identifies two states more prominently. Panel **b** highlights the regions assigned a higher consensus error for each method. In both methods, the most unstable particles tend to organize in the periphery of the consensus conformational landscape, with the main regions of the landscape being more stable. SWD, sliced Wasserstein distance.

are found in the periphery of the landscape, which are also regions with a lower density than the central cloud. These differences are further highlighted in Extended Data Fig. 2. It should be noted that the consensus landscape has three dimensions as specified when defining the network, although Fig. 3 shows only a two-dimensional projection. Thus, some of the highest error points in HetSIREN seem to be located within the landscape, although the visualization in three dimensions reveals that they are also located in the periphery.

To further support the results observed from the inspection of the consensus space, in Extended Data Fig. 3 we show the error histograms computed from the representation error obtained from the comparison of the input spaces against their analogous decoded spaces. Similar to the previous sections, each consensus space (that is, the points in the consensus space coming from the Zernike3D and HetSIREN landscapes) can be decoded toward each input space. Therefore, we have three decoded spaces analogous to Zernike3D, three to HetSIREN in reconstruction mode and three to HetSIREN in refinement mode. With the previous histograms, Extended Data Fig. 3 also includes the consensus error derived directly from the common consensus space.

As expected, the two HetSIREN runs are more similar, meaning that the two executions were relatively stable. This also means that the HetSIREN operation mode has an inconsiderable effect in determining the conformational landscape in this specific case. It is also possible to detect how Zernike3D errors tend to deviate more from HetSIREN and the effect arising from the differences in the type of heterogeneity that both methods can detect. From the histograms, one can quickly identify and interactively determine the particles to exclude from future analysis, as they accurately summarize how reliable the heterogeneity estimation was for every particle.

To better understand the source of the differences detected by FlexConsensus between the two HetSIREN executions and Zernike3D, an additional experiment was conducted to analyze at volume level the conformations arising from the landscape regions detected to have the most substantial errors. To that end, a subset of the HetSIREN and Zernike3D landscapes was extracted based on the indices obtained from the images, which were estimated to have the most substantial error based on the consensus. The subset spaces were then clustered to get 10 different representatives of the conformations, which were used to decode their associated conformations with HetSIREN and Zernike3D. The different conformations obtained are presented in Supplementary Video 3. As seen from comparing the states recovered by both methods, HetSIREN tends to detect motions with lower amplitude and a strong compositional component. In comparison, Zernike3D estimates continuous motions with a larger amplitude. This result suggests that the consensus differences are not related to 'wrong' images but to estimation differences between the two methods.

### Consensus results on SARS-CoV-2 D614G dataset

To further assess the capabilities of FlexConsensus under different experimental conditions, we evaluated the method with the SARS-CoV-2 D614G spike using a dataset obtained from Sriram Subramanian's laboratory (University of British Columbia). This protein

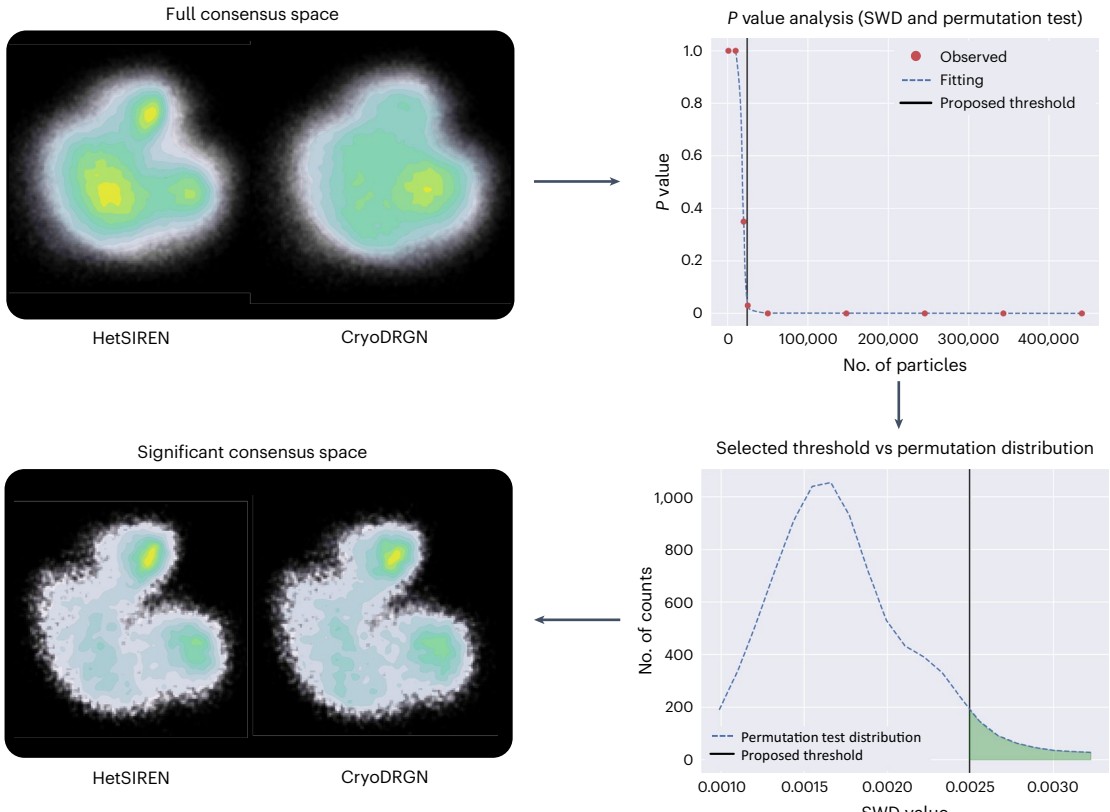

**Fig. 5 | Scheme of the stabilization process to improve the state distribution reliability of the shared FlexConsensus landscape.** The process relies on the hypothesis that HetSIREN and CryoDRGN have found the same distribution of conformational states. From this assumption, it is possible to determine the threshold verifying the hypothesis (which corresponds to a P value of 0.05 from the one-sided test), allowing one to obtain a significant consensus space with a more accurate and reliable distribution of states.

is well characterized and exhibits a wide range of motions, primarily affecting the prefusion state, the RBD and the N-terminal domains. The RBD transitions are particularly interesting among the motions the spike undergoes due to their localized and highly dynamic nature.

Similar to the pipeline followed in the previous section, the experimental dataset was preprocessed within Scipion[12], leading to 440,000 particle images with CTF and angular information. The structural variability captured by the particles was then approximated with two different software: HetSIREN[5] and CryoDRGN[2]. In contrast to Zernike3D discussed in the previous section, both HetSIREN and CryoDRGN follow the heterogeneous reconstruction approximation to extract the conformational landscape from a set of images. Their conformational landscapes should be more comparable because they follow a similar approximation to solve the structural heterogeneity problem.

It is worth mentioning that HetSIREN can further refine the poses of the images while estimating the conformational states, in contrast to CryoDRGN. However, it was decided to turn off this feature in HetSIREN so that the two methods behave similarly when estimating their conformational latent spaces, simplifying the data analysis.

Again, the analysis will follow the same terminology introduced in the previous experiment: 'X subspace', the space determining the location of the points coming from input space X in the common consensus space, and 'common consensus space', composed of all the subspaces.

The reliability of the two independently estimated landscapes was analyzed by FlexConsensus, leading to the consensus landscapes presented in Fig. 4. Fig. 4a includes the common consensus landscape in grays and the HetSIREN and CryoDRGN subspaces colored on top (note that this two-dimensional image of consensus space corresponds to a selected view of the computed 3D space; full access to 3D space can be done through Flexutils, as described later on in the main text).

An initial inspection of these representations reveals three central structural regions corresponding to the RBDs in three-down, one-up and two-up states, correctly identified by both methods. However, HetSIREN concentrates more on the three-down and one-up states, unlike CryoDRGN, which focuses more on the two-up and less on the three-down. In addition, Fig. 4b shows the location of the images estimated to have a more substantial consensus error based on the structural variability estimation from HetSIREN and CryoDRGN. In both cases, these particles are located in the periphery of the consensus landscape, this arrangement being a more prominent feature in the case of HetSIREN. Particles located in the periphery are usually associated with states estimated wrongly due to bad particle images or random estimation errors. Again, it is important to note that the consensus landscape lives in a 3D space. Thus, some of the highest error points shown in Fig. 4b appear to be in the interior of the landscape. However, the visualization in three dimensions reveals that they are also located in the periphery. Overall, the presence of these periphery particles is not important compared to the inner region of the consensus landscapes, where a more thorough analysis can be performed to properly describe whether the differences in the distribution of state estimated by different methods are considerable.

To more quantitatively assess the reliability of the structural states estimated by HetSIREN and CryoDRGN, we evaluated the representation error between the input spaces and those decoded from the consensus space. Four spaces were decoded, obtained when forwarding the HetSIREN and CryoDRGN consensus spaces through the decoders responsible for generating the original two spaces from the consensus. These four spaces were then used to compute the representation errors represented as histograms in Extended Data Fig. 4, and the consensus error was measured directly on the consensus space. The histogram

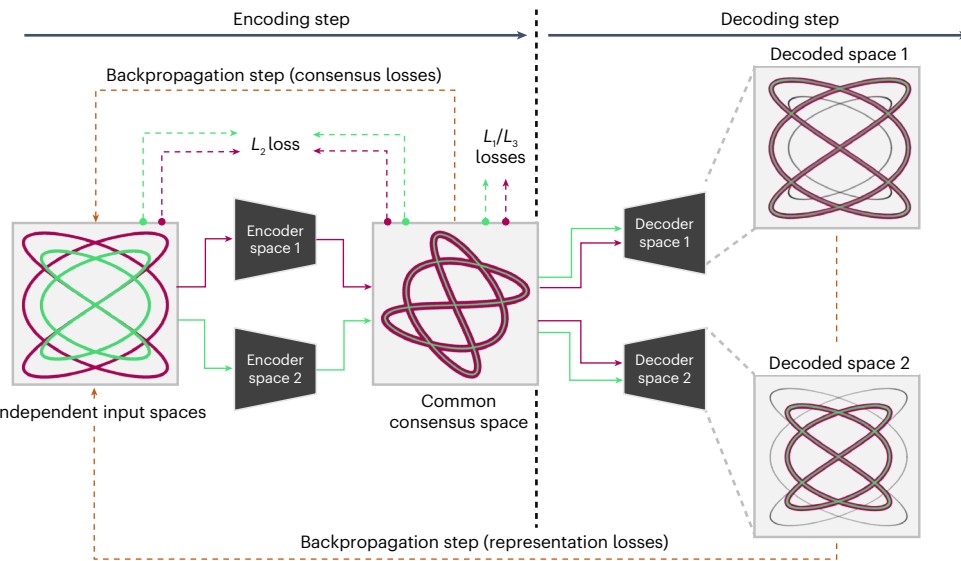

**Fig. 6 | Scheme of the FlexConsensus training workflow.** A set of independent spaces is fed to different encoders, generating an independent representation of the initial spaces on a common consensus space. All encoded consensus spaces are then forwarded to the first decoder, which is responsible for transforming the consensus spaces into a space as similar as possible to the first input space. A representation loss is evaluated and backpropagated through the network at this step. The previous decoded step is repeated sequentially for all the decoders, allowing the training of all the weights of the multi-autoencoder network.

analysis reveals a slight discrepancy between the two methods, as the error distributions between the spaces decoded from each consensus space do not fully overlap. This discrepancy suggests that the relative population of states estimated by CryoDRGN and HetSIREN is unequal, even though both methods correctly detect the main conformational states in the dataset. This is further supported when visually inspecting the consensus spaces included in Fig. 4, as it can be seen that the overall distribution of the particles over the three main spike states is not similar in the two methods.

It should be noted that the relative error scales will depend on the relative scale of the input space, making the comparison and interpretation of the error histograms difficult. In general, the scale of the conformational space for two different methods might not be similar, meaning that the latent vectors might be larger or smaller depending on how the method is designed. The differences in the scale of the latent vectors are transmitted through the decoders in FlexConsensus to find a mapping that is as close as possible to a given input. Therefore, the relative differences in scale are reflected in the relative representation error computed for a given input space. Thus, we applied the normalization described in the Supplementary Information to simplify the comparison of the error histograms presented in Extended Data Fig. 4 and improve their interpretability. This normalization makes it easier to understand the relative size of the errors while constraining them to the same range.

Similar to the previous experiments performed with CryoBench datasets, we filtered the consensus space obtained from the SARS-CoV-2 dataset to keep only those images estimated to have a significantly similar conformation. The results obtained for this analysis are summarized in Fig. 5.

When comparing the results obtained from the two analyses presented in Fig. 5 and Extended Data Fig. 4, it is possible to see that both approaches detect that the distribution of states identified by HetSIREN and CryoDRGN was incompatible in an important fraction of the images. These results suggest the importance of validating the landscapes estimated by a given algorithm to detect these inconsistencies.

However, it is interesting to understand whether the filtered FlexConsensus space yields more meaningful landscape populations. To that end, we conducted a 3D classification of the original images into several classes, which were merged according to the RBD conformational state that they represent. Because both HetSIREN and CryoDRGN suggest a clear tendency toward the presence of three well-differentiated conformational states, the populations of the 3D classification will be considered as the 'ground truth' populations as they are obtained from a standard approach in the field.

In the case of the landscapes, Gaussian mixture-based clustering was used to compute the populations. To that end, three Gaussian components were estimated to cluster the spaces to get the relative populations to compare against the 3D classification ones. It should be noted that the populations were directly measured in the consensus subspaces of HetSIREN and CryoDRGN, which were obtained using FlexConsensus.

The relative populations obtained were as follows:

- 3D classification (Relion[17]): three down (20%), one up (60%) and two up (20%)
- HetSIREN (FlexConsensus): three down (22%), one up (50%) and two up (28%)
- CryoDRGN (FlexConsensus): three down (27%), one up (35%) and two up (38%)
- FlexConsensus filtered landscape: three down (26%), one up (41%) and two up (31%)

By comparing the different populations against the 3D classification, it is possible to see how HetSIREN tends to be more conservative, estimating a set of populations that follow a similar trend to the 3D classification ones (that is, a larger number of one-up states compared to the other two). By contrast, the trend of CryoDRGN is to give a larger population to the two-up states. FlexConsensus filtered landscape tends to normalize the populations to a value between the two original methods, leading to a new set resembling the behavior seen in the classification results.

A question remains on analyzing the characteristics of those images estimated to have a large consensus error. It is important to understand whether they correspond to 'bad' particles or, instead, their position in conformational space arises from estimation errors from the heterogeneity methods. Therefore, an additional experiment was conducted to compare the reconstructions obtained from those images estimated to have a consensus error larger than a given

threshold. The results obtained in this analysis are summarized in Extended Data Fig. 5. As shown in the figure, and as expected, these particles are distributed differently in the HetSIREN and CryoDRGN consensus spaces. In CryoDRGN, the images are placed close to the two-up region, whereas HetSIREN tends to distribute them along the one-up and three-down states. By contrast, the volume reconstructed from these images yields a clear one-up state, indicating that the quality of these images was not intrinsically bad but that, for yet unknown reasons, both HetSIREN and CryoDRGN did not properly position these images in conformational space close to other images in similar states of RBD opening. Consequently, although the conceptual distinction between images with large consensus errors being either intrinsically bad or badly estimated by a method is very important, it seems that we do not have a clear way to distinguish between these two options.

## Discussion

The cryo-EM community's great interest in the emerging field of conformational variability analysis is reflected in the increasing number of advanced methods published in recent years. However, the interest in these tools raises a new challenge in comparing the estimations of different techniques to assess their stability, reliability and accuracy. Moreover, the diverse nature of all these new methods makes comparing their results more complex, making it even more difficult to find robust approaches to accurately defining a valid consensus analysis.

To allow a better understanding of conformational landscapes, in this work we discuss a new method to overcome the challenges arising from the comparison of different heterogeneity algorithms, leading to a whole range of possibilities to extract more reliable and accurate conformational states from independent conformational landscapes. Our new FlexConsensus approach relies on a multi-autoencoder architecture specifically designed to robustly identify commonalities and differences among multiple conformational landscapes, ultimately defining a common consensus landscape with enhanced interpretability. From the consensus landscape, the network can automatically derive a consensus error metric for every particle in the conformational landscape, which can be used either to explicitly determine for any given image in a dataset its structural state's reliability across various heterogeneity methods, determining subsets that can be confidently analyzed, or to analyze specific subsets of particles highlighted only by particular algorithms.

## Online content

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

## Methods

This section describes the FlexConsensus multi-autoencoder architecture and the training strategy followed to allow the network to learn meaningful consensus spaces and consensus metrics from input conformational spaces. After the architecture's discussion, we include the description of the filtering method applied in this paper and a brief discussion on how to access and integrate FlexConsensus in a workflow.

### FlexConsensus multi-autoencoder architecture

FlexConsensus architecture consists of a multi-autoencoder architecture with a variable number of encoders and decoders determined by the total number of inputs. Each input corresponds to a conformational landscape that the consensus network should consider, which may come from a different algorithm with its own characteristics. An architecture scheme is provided in Fig. 6.

Each input conformational landscape is forwarded through its encoder, which is composed of three fully connected layers with 1,024 neurons each and rectified linear unit (ReLU) activation. The output layer of every encoder is then fed to a standard linearly activated latent space layer of a variable number of neurons that the user can choose. This bottleneck will become the common conformational space after the network is trained.

Lastly, each encoded data are forwarded through the corresponding decoder, trying to restore the original conformational space based on the information of the common latent space. Similar to the encoders, each decoder comprises a set of three fully connected layers with 1,024 neurons each and ReLU activation, followed by a final layer with the same dimension as the corresponding initial conformational space and linear activation.

The network is trained using an Adam optimizer. By default, the learning rate is set to $10^{-5}$, and the batch size is 1,024. However, users can modify these hyperparameters to fit their needs.

As a reference, training the network takes approximately 16 seconds per epoch on an RTX Ada 6000 generation GPU, using default hyperparameters and two input spaces with 450,000 points each; VRAM consumption is approximately 2.6 GB.

### Ensuring proper merging of conformational spaces

One of the main challenges in training a consensus network is correctly driving the network to merge spaces of varying characteristics in the same region of the latent space. Due to the properties of autoencoders, a simple mean square error between the original and predicted spaces results in a latent space where different distributions are well separated. Although this will help decrease the network's representation error, it is a highly undesirable effect, as it completely obscures the understanding of the intrinsic relations among the distributions of the input spaces.

A possible approach to solving the previous problem is to bring the latent space distributions close by, ensuring that their distances are as small as possible. This can be easily achieved by minimizing the mean square error between all possible distribution pairs. In this way, the network is forced to learn a more meaningful representation compared to the original uncontiguous case:

$$L_1 = \min \sum_{n=1}^{N} \sum_{m=1}^{N} \sum_{i} \left\| E_n\left(\mathbf{x}_i^n\right) - E_m\left(\mathbf{x}_i^m\right) \right\|^2 \tag{1}$$

being $N$, the total number of input spaces; $E_n$ and $E_m$ are the operators responsible for encoding a given space into the common consensus space; and $i$ is an index that covers all input points. $\mathbf{x}_i^n$ is the representation of the $i$-th image latent vector in the $n$ latent space.

However, the previous cost is not sufficient in our experiments, primarily due to the effect of the representation error of the decoder, which drove the result away from a fully merged consensus space. Therefore, an additional restriction is required to properly ensure

that the local structure of the original conformational spaces is preserved as much as possible in the predicted consensus space. To that end, we propose comparing the distances between the original and predicted latent spaces. The self-distance matrices of every input and latent space batch are computed and compared on a pairwise basis following a similar approach to the previous cost. The minimization of the all-to-all pairwise mean square error among the self-distance matrices allows the network to learn to keep the original structure of the conformational spaces as consistent as possible. This is similar to the objective of Shannon mapping. Therefore, the Shannon mapping cost function can be written as:

$$L_2 = \min \sum_{n=1}^{N} \left( \frac{\sum_{i,j} \frac{(d_{ij}(X^n) - d_{ij}(E_n(X^n)))^2}{d_{ij}(X^n)}}{\sum_{i,j} d_{ij}(X^n)} \right) \tag{2}$$

being $d_{ij}(X)$ the $ij$-th entry of the self-distance matrix computed from a set of coordinates $X$. The self-distance matrix is computed as the all-versus-all distance among the coordinates in the set.

In addition, a third cost is included in the total cost function to ensure that the distance distributions among the different consensus spaces are similar. To that end, we compute the self-pairwise distance matrix of the other encoded consensus spaces. The distance matrix is then calculated, and its entries are considered random samples from a distance distribution. We then add another regularization term based on the Wasserstein distance between two different distributions:

$$L_3 = \min \sum_{n=1}^{N} \sum_{m=1}^{N} W(D(E_n(X^n)), D(E_m(X^m))) \tag{3}$$

being $D$, the operator that computes the distance matrix, and $W$, the function that computes the Wasserstein distance between two distance distributions.

Combining the previous three costs at the latent space level in our experiments led to a good representation of a continuous consensus space. In addition to the last three cost functions, a standard autoencoder representation error between the original and predicted conformational spaces based on the consensus space is also included. Because every conformational space might be affected by different estimation errors, only those regions consistently placed in the consensus space will have a low representation error. Therefore, the representation error provides a good measurement of the stability of the estimations, which can be used in conjunction with the consensus space to detect or filter out those regions in the conformational spaces that might have been estimated less reliably based on the agreement of all the spaces. The contribution of the previous regularizers in the overall cost function is summarized as follows:

$$L = \sum_{n=1}^{N} \sum_{m=1}^{N} \sum_{i} \left\| \mathbf{x}_i^n - D_n\left(E_m\left(\mathbf{x}_i^m\right)\right) \right\|^2 + L_1 + L_2 + L_3 \tag{4}$$

The first term represents the representation loss obtained by comparing the inputs against the decoded outputs, and $D_n$ is the corresponding decoder responsible for transforming a given consensus space into the input space $n$.

### FlexConsensus training strategy

The multi-encoder architecture proposed in FlexConensus introduces extra flexibility in the training step, depending on the path to backpropagate the gradients computed during every forward pass through each autoencoder.

The workflow followed to train FlexConsensus is shown in Fig. 6. In the simplified case proposed in the figure, two independent input spaces are fed to their corresponding encoders, generating a different representation in a shared latent space. The first step includes encoding

both input spaces, leading to two independent representations in the consensus space. Once the consensus spaces are generated, they are forwarded sequentially through the different decoders, triggering independent backpropagation steps for every decoder. We found the sequential training of the decoders to be more stable than training the whole network on a single backpropagation step, leading to a faster and more accurate convergence.

Once the network is trained, the corresponding encoder can recreate the complete consensus space and the location of a specific input in that space. In addition, the different encoder/decoder combinations can be used to convert to different input spaces and measure the consensus error estimated for a given input.

### Consensus set

Given the mapping of all points from the input spaces to a shared latent space, a natural question arises: which images indeed agree with one another? As illustrated in Fig. 6, we can calculate the distance between the projected points of space 1 and their corresponding projections from space 2 in the latent space. Although this approach can be extended to more than two spaces, we simplify the explanation by focusing on two spaces. We rank all points based on their average distance to points in the other space(s). Intuitively, the closest points represent agreement, whereas the most distant points indicate disagreement.

Using the top $K$ particles from this list, we compare the distributions of their projections from space 1 and space 2. This comparison is based on the Wasserstein distance between the two sets of points. We approximate the Wasserstein distance in $k$ dimensions to accelerate the calculations using the one-dimensional Wasserstein distance of the points projected onto $\kappa$ random unit vectors[18]. We denote this observed distance as $d_{obs}$. To assess its significance, we compare $d_{obs}$ to the distribution of distances obtained by randomizing the labels 'space 1' and 'space 2' (performing 100 randomizations). The proportion of randomized distances smaller than or equal to $d_{obs}$ gives the $P$ value, representing the probability of observing such a distance under random labeling. We expect that points with a small representation error in the latent space correspond to indistinguishable distributions of latent points from space 1 and space 2. However, as the number of particles $K$ increases, a threshold is reached where the two distributions become distinguishable, indicating that the two sets of points can no longer be considered equivalent. This threshold is defined as the point where the $P$ value falls below 0.05. The value of $K$ at which this occurs establishes the set of consensus images.

### FlexConsensus as part of heterogeneity analysis workflows—the case of integration inside Scipion

Flexibility analysis is becoming progressively a standard practice in the cryo-EM field, promoted by the new emerging tools that can extract more accurately the structural information captured in the experimental images. Therefore, it is common for experimentalists to analyze the flexibility of their biomolecules with these tools. However, they are restricted in the depth of interpretation of the results reported by any given method because of the lack of validation tools and the difficulty in performing a joint analysis of the results of different methods. Our method, FlexConsensus, aims to be inserted at this point in the analysis, allowing the integration and comparison of the results yielded by different algorithms to validate whether the conformational spaces they report agree.

The FlexConsensus algorithm has been integrated into Scipion as a protocol included in the Scipion Flexibility Hub[19] and accessible through the Flexutils Scipion plugin. The protocol provides a graphical interface, allowing one to easily customize the network's hyperparameters or select the neural network's inputs.

Inside Scipion, it is possible to easily execute popular heterogeneity analysis methods such as CryoDRGN[2], 3DFlex[7] or HetSIREN[5], among others. The inputs needed to execute these algorithms are also organized in a form similar to the FlexConsensus case. The particles required to execute them can be imported into Scipion or processed inside by combining cryo-EM software such as Relion[17] or CryoSPARC[15]. The conversions needed to adapt the outputs and inputs to different software are internally handled by Scipion, making the transition among packages smoother. These concepts also apply to the heterogeneity software integrated into Scipion and FlexConsenus.

The predictions obtained after training FlexConsensus are also automatically registered inside Scipion, allowing it to communicate with the integrated visualization, analysis and subset generation tools and giving the user greater flexibility to manipulate their data according to their specific needs.

For example, users can access the 3D space annotator to view the consensus space in three dimensions, allowing them to select any combination of axes/dimensions to inspect and interact with it. The annotation tool also enables the recovery of real-time confirmations from any region of the consensus space to visualize them inside the annotator or using the direct connection with ChimeraX[20]. To extract the previous conformations, the annotator relies on the specific method that generated the latent space being analyzed to recover the same state seen by that method for any location in the conformational space. This avoids the need to reconstruct with a neighborhood of particles surrounding a given point in the conformational space.

Additionally, it is possible to extract the particles associated with a given region interactively and register them inside Scipion, allowing their subsequent processing in the workflow to analyze them further.

### Reporting summary

Further information on research design is available in the Nature Portfolio Reporting Summary linked to this article.

### Data availability

CryoBench datasets are publicly available at https://cryobench.cs.princeton.edu/. The 80S ribosome experimental dataset analyzed in the paper is available in EMPIAR under entry 10028 (https://doi.org/10.6019/EMPIAR-10028). The SARS-CoV-2 D614 spike dataset will be published in a different work.

### Code availability

The FlexConsensus algorithm is available through Scipion 3.0 (ref. 12) under the plugins and scipion-em-flexutils[21] (https://github.com/scipion-em/scipion-em-flexutils) and the package Flexutils-Toolkit[22] (https://github.com/I2PC/Flexutils-Toolkit). The protocol corresponding to the algorithm described in this paper is flexutils - train - FlexConsensus and flexutils - interactive consensus - FlexConsensus. Tutorials on how to set up and use FlexConsensus are provided on the following webpage: https://scipion-em.github.io/docs/release-3.0.0/docs/user/tutorials/flexibilityHub/main_page.html#tutorials. In addition, we provide an interactive Scipion workflow visualization showing an example of how to execute FlexConsensus here: https://scipion.i2pc.es//cryoemworkflowviewer/workflow/aac223841371a67510e9eab16ac8870246b30d68.

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

## Acknowledgements

The authors acknowledge the financial support from the Ministry of Science, Innovation, and Universities (BDNS no. 716450) to Instruct-ES as part of the Spanish participation in Instruct-ERIC; the European Strategic Infrastructure Project (ESFRI) in the area of Structural Biology, grant PID2022-136594NB-I00 (J.M.C. and C.O.S.S.), funded by MICIU /AEI/10.13039/501100011033/ and 'ERDF A way of making Europe' by the European Union; the Spanish State Research Agency AEI/10.13039/501100011033, through the Severo Ochoa Programme for Centres of Excellence in R&D (CEX2023-001386-S) (J.M.C. and C.O.S.S.); Comunidad Autónoma de Madrid through grant S2022/BMD-7232 (J.M.C. and C.O.S.S.); the European Union and Horizon 2020 through grant HighResCells (ERC - 2018 - SyG, Proposal: 810057) (J.M.C. and C.O.S.S.); and the European Union and Horizon Europe through grant Fragment Screen Proposal: 101094131 (J.M.C. and C.O.S.S.). We thank S. Subramaniam for providing us with the data processed and presented in this work and for his support and feedback.

## Author contributions

D.H. developed and tested the FlexConsensus method presented throughout the paper. D.H. wrote the manuscript. C.P.M. helped with the data preprocessing. C.O.S.S. and J.M.C. jointly supervised this work.

## Competing interests

The authors declare no competing interests.

## Additional information

**Extended data** is available for this paper at https://doi.org/10.1038/s41592-025-02841-w.

**Correspondence and requests for materials** should be addressed to David Herreros.

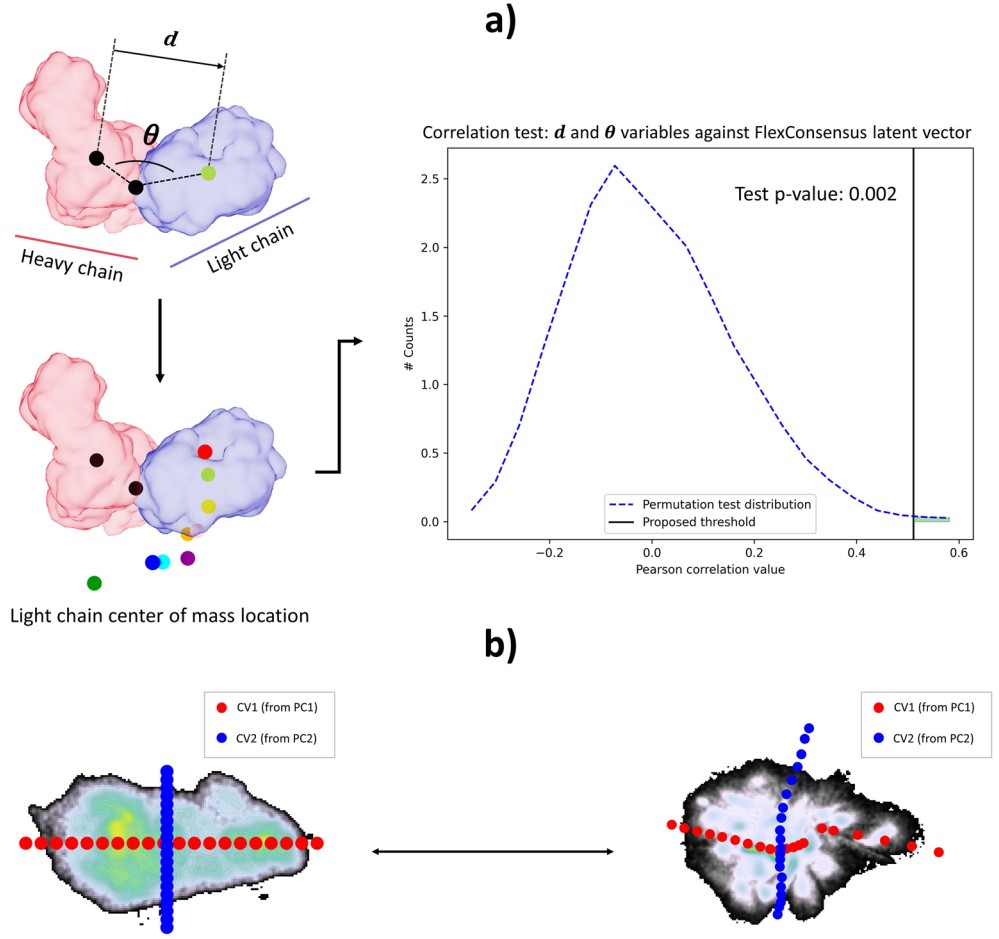

**Extended Data Fig. 1 | Evaluation of CryoBench dataset results for the IgG-RL and MDSpike datasets.** Panel **a**) shows the evaluation of the correlation between the FlexConsensus latent space vectors obtained for the IgG-RL dataset from CryoBench[11] and two internal variables ($d$ and $\theta$) designed to characterize simulated disordered motions (the Mantel test used to evaluate the significance of this correlation was two-sided, yielding a p-value of 0.002). Panel **b**) shows the comparison of the location of the collective variables measured from the ground truth energy landscape of the MDSpike dataset from CryoBench[11] in its original energy landscape and FlexConsensus space. Three landscapes were input to the consensus: the ground truth energy landscape, the conformational landscape of HetSIREN, and the conformational landscape of CryoDRGN. The results highlight the correlation of the original collective variables (CV1 and CV2) with the consensus space, which are almost aligned with the consensus space's principal components (PC).

**Consensus differences (strong compositional component)**

Consensus space (Zernike3D)    Consensus space (HetSIREN rec)    Consensus space (HetSIREN ref)

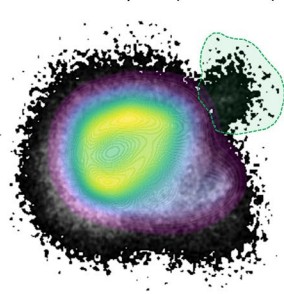 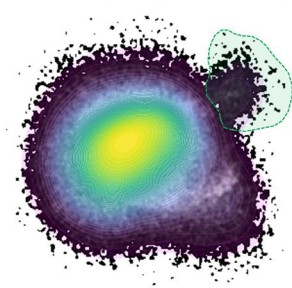 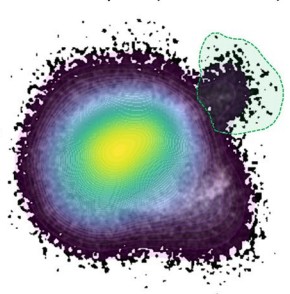

**Consensus differences (wrongly reconstructed particles)**

Consensus space (Zernike3D)    Consensus space (HetSIREN rec)    Consensus space (HetSIREN ref)

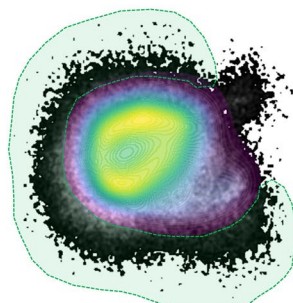 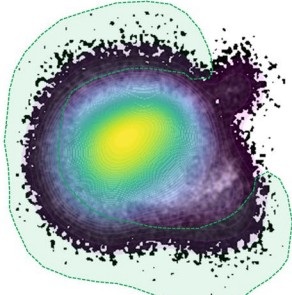 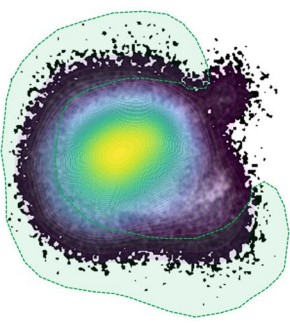

**Extended Data Fig. 2 | Highlight of the main differences found in the consensus landscape estimated for the EMPIAR 10028 dataset.** The top panel shows the regions where Zernike3D and HetSIREN disagree due to the presence of a strong compositional change that Zernike3D cannot capture. The bottom panel shows a strong difference in the periphery of the landscape, mainly associated with wrongly estimated conformations with increasing noise in the case of HetSIREN. Since Zernike3D cannot add noise to the conformation it estimates, its periphery is closer to the central region of the consensus space, as shown in the Figure.

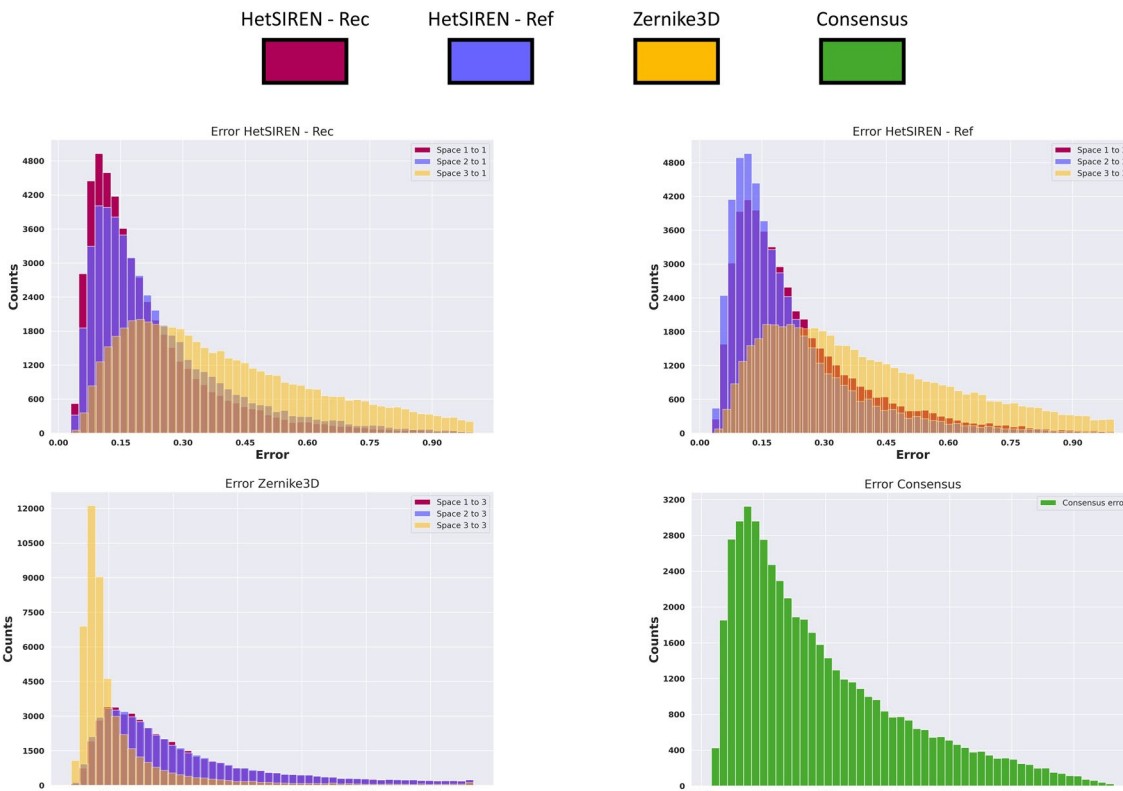

**Extended Data Fig. 3 | FlexConsensus error histograms computed for the EMPIAR 10028 dataset.** The error metric corresponds to the standard Root Mean Squared Error between every original space and the spaces decoded by FlexConsensus from the consensus space, as well as the consensus error derived from the consensus space. The three images presented show a similar error tendency for all the decoders: the two HetSIREN executions remain more similar, meaning that the two independent estimations have similar characteristics and are reliable for most particles. In contrast, Zernike3D deviates more from HetSIREN, an effect mainly occurring due to the differences in the approximation followed by each method and the differences in the type of variability they can estimate. Apart from these differences, one can see that the distribution of the errors is similar in the three cases, and it allows us to easily identify and exclude those particles on the right tail of the distributions, which are the ones estimated with lower reliability based on these three executions.

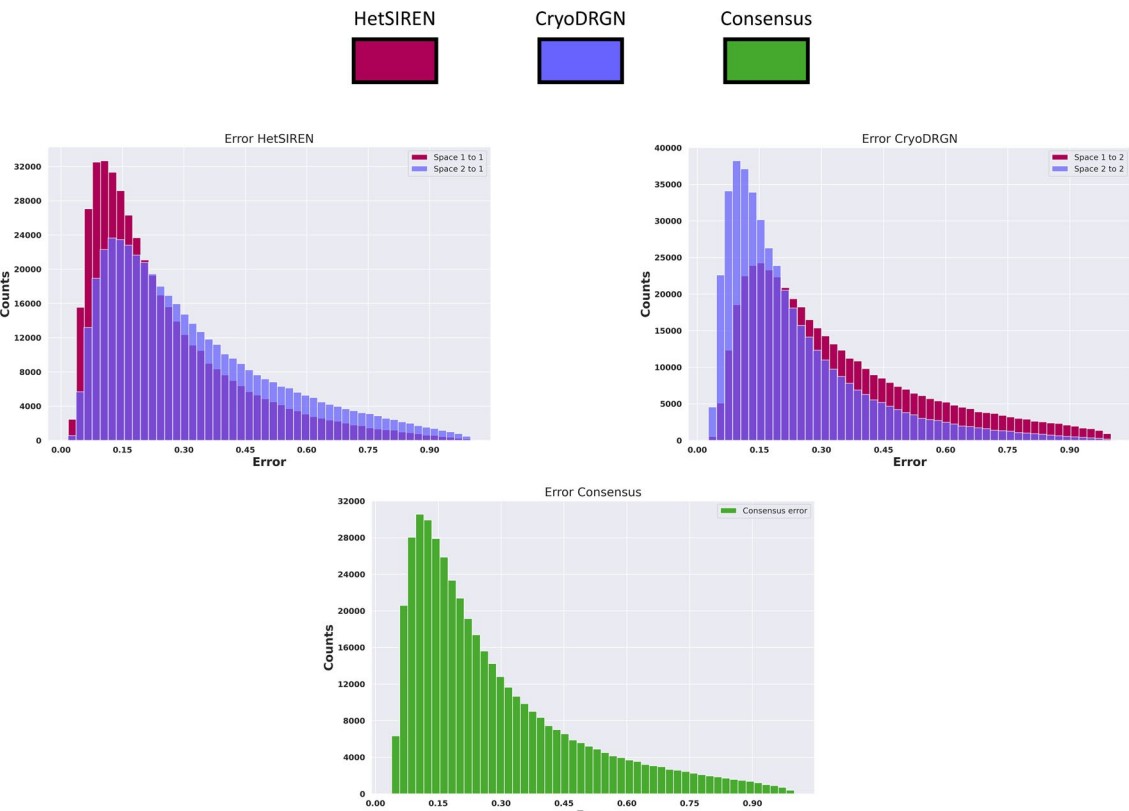

**Extended Data Fig. 4 | FlexConsensus error histograms computed for the SARS-CoV-2 D614G spike.** The error metric corresponds to the standard Root Mean Squared Error between every original space and the spaces decoded by FlexConsensus from the consensus space, as well as the consensus error derived from the consensus space. The histograms reveal that the heterogeneity estimations yielded by HetSIREN and CryoDRGN suffer from a slight deviation, suggesting that the landscapes estimated by both methods might not follow the same distribution of states.

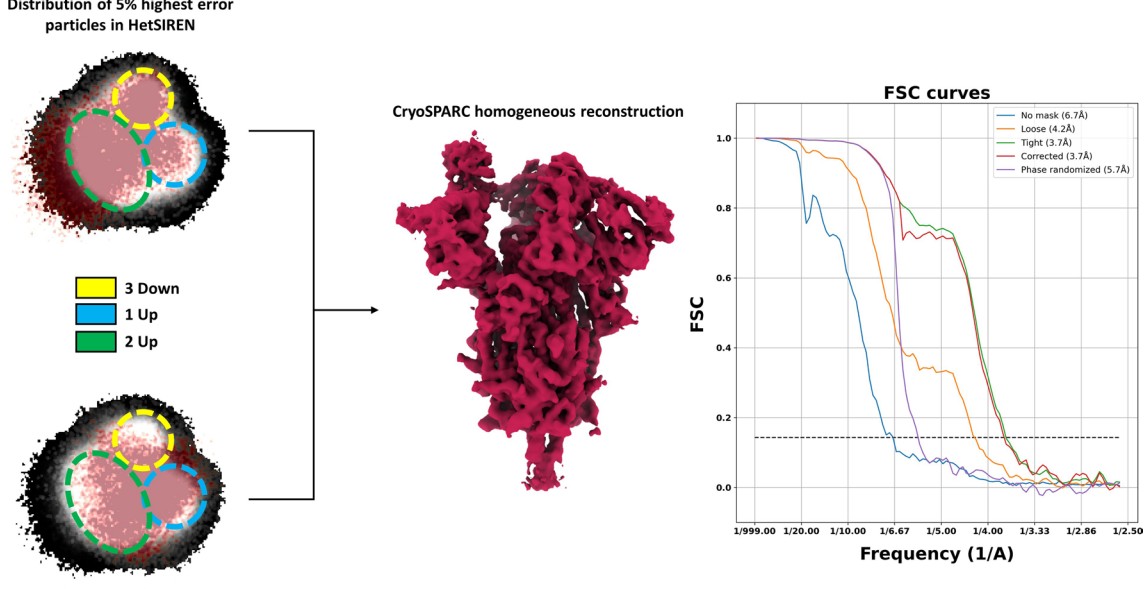

**Extended Data Fig. 5 | Analysis of those images estimated to have a larger consensus error in the SARS-CoV-2 dataset.** Reconstruction obtained from the SARS-CoV-2 dataset, considering only those images estimated to be within the 5% largest consensus error (around 21k particles from the original 440k). The landscapes show the distribution of these images in the consensus space: For CryoDRGN, most of these images are located in the two-up state region, while in HetSIREN, they are distributed along the regions corresponding to the one-up and three-down states. In contrast, the reconstruction calculated directly from these particles corresponds to a clear one-up state, indicating that the two heterogeneity methods wrongly assigned these images to regions of the conformational landscape.

# Reporting Summary

## Statistics

For all statistical analyses, confirm that the following items are present in the figure legend, table legend, main text, or Methods section.

| n/a | Confirmed | |
|---|---|---|
| ☐ | ☒ | The exact sample size (*n*) for each experimental group/condition, given as a discrete number and unit of measurement |
| ☒ | ☐ | A statement on whether measurements were taken from distinct samples or whether the same sample was measured repeatedly |
| ☐ | ☒ | The statistical test(s) used AND whether they are one- or two-sided *Only common tests should be described solely by name; describe more complex techniques in the Methods section.* |
| ☒ | ☐ | A description of all covariates tested |
| ☒ | ☐ | A description of any assumptions or corrections, such as tests of normality and adjustment for multiple comparisons |
| ☐ | ☒ | A full description of the statistical parameters including central tendency (e.g. means) or other basic estimates (e.g. regression coefficient) AND variation (e.g. standard deviation) or associated estimates of uncertainty (e.g. confidence intervals) |
| ☐ | ☒ | For null hypothesis testing, the test statistic (e.g. *F*, *t*, *r*) with confidence intervals, effect sizes, degrees of freedom and *P* value noted *Give P values as exact values whenever suitable.* |
| ☒ | ☐ | For Bayesian analysis, information on the choice of priors and Markov chain Monte Carlo settings |
| ☒ | ☐ | For hierarchical and complex designs, identification of the appropriate level for tests and full reporting of outcomes |
| ☒ | ☐ | Estimates of effect sizes (e.g. Cohen's *d*, Pearson's *r*), indicating how they were calculated |

*Our web collection on statistics for biologists contains articles on many of the points above.*

## Software and code

Policy information about availability of computer code

| Data collection | No data was collected for this manuscript. |
|---|---|
| Data analysis | All the data was analyzed with Scipion 3.8.0 software package. Inside Scipion, CryoDRGN v3.4.0 was also used to estimate the conformational landscapes analyzed in the manuscript, as well as Flexutils plugin v3.3.0.

CryoSPARC 4.5.1, Relion 4.0, and Xmipp 3.24.12.0 packages were also used to preprocess the data.

FlexConsensus algorithm is available through Scipion 3.0 under the plugins and "scipion-em-flexutils" (https://github.com/scipion-em/scipion-em-flexutils) and the package "Flexutils-Toolkit" (https://github.com/I2PC/Flexutils-Toolkit). |

For manuscripts utilizing custom algorithms or software that are central to the research but not yet described in published literature, software must be made available to editors and reviewers. We strongly encourage code deposition in a community repository (e.g. GitHub). See the Nature Portfolio guidelines for submitting code & software for further information.

## Data

Policy information about availability of data

All manuscripts must include a data availability statement. This statement should provide the following information, where applicable:

- Accession codes, unique identifiers, or web links for publicly available datasets
- A description of any restrictions on data availability
- For clinical datasets or third party data, please ensure that the statement adheres to our policy

> The experimental datasets analyzed in the manuscript are available in EMPIAR under the entries: 10028 [https://doi.org/10.6019/EMPIAR-10028]. The e SARS-CoV-2 D614 spike dataset will be published in a different work in the future. CryoBench datasets are publicly available through the following link https://cryobench.cs.princeton.edu/

## Research involving human participants, their data, or biological material

Policy information about studies with human participants or human data. See also policy information about sex, gender (identity/presentation), and sexual orientation and race, ethnicity and racism.

| | |
|---|---|
| Reporting on sex and gender | Not applicable |
| Reporting on race, ethnicity, or other socially relevant groupings | Not applicable |
| Population characteristics | Not applicable |
| Recruitment | Not applicable |
| Ethics oversight | Not applicable |

Note that full information on the approval of the study protocol must also be provided in the manuscript.

# Field-specific reporting

Please select the one below that is the best fit for your research. If you are not sure, read the appropriate sections before making your selection.

☒ Life sciences  ☐ Behavioural & social sciences  ☐ Ecological, evolutionary & environmental sciences

For a reference copy of the document with all sections, see nature.com/documents/nr-reporting-summary-flat.pdf

# Life sciences study design

All studies must disclose on these points even when the disclosure is negative.

| | |
|---|---|
| Sample size | This information is not relevant for image processing data. |
| Data exclusions | No data was excluded from the analyses. |
| Replication | All experiments were repeated more than once to ensure that the results and conclusions derived from them were reproducible. |
| Randomization | This is not relevant to this study, as the datasets are independent and cannot be mixed into groups. |
| Blinding | This is not relevant for this study, as the datasets are not blind during their analysis. |

# Reporting for specific materials, systems and methods

We require information from authors about some types of materials, experimental systems and methods used in many studies. Here, indicate whether each material, system or method listed is relevant to your study. If you are not sure if a list item applies to your research, read the appropriate section before selecting a response.

## Materials & experimental systems

| n/a | Involved in the study |
|---|---|
| ☒ ☐ | Antibodies |
| ☒ ☐ | Eukaryotic cell lines |
| ☒ ☐ | Palaeontology and archaeology |
| ☒ ☐ | Animals and other organisms |
| ☒ ☐ | Clinical data |
| ☒ ☐ | Dual use research of concern |
| ☒ ☐ | Plants |

## Methods

| n/a | Involved in the study |
|---|---|
| ☒ ☐ | ChIP-seq |
| ☒ ☐ | Flow cytometry |
| ☒ ☐ | MRI-based neuroimaging |

## Plants

| Seed stocks | Not applicable |
|---|---|
| Novel plant genotypes | Not applicable |
| Authentication | Not applicable |

