## [Peer Review File · Nature Methods]

Merging conformational landscapes in a single consensus space with FlexConsensus algorithm

Corresponding Author: Dr David Herreros Calero

Version 0:

Decision Letter:

18th Mar 2025

Dear Dr. Herreros,

Your Article, "Merging conformational landscapes in a single consensus space with FlexConsensus algorithm", has now been seen by 3 reviewers.

As you will see from their comments below, although the reviewers find your work to be conceptually of considerable potential interest, they have raised a number of important technical concerns. We are interested in the possibility of publishing your paper in Nature Methods, but would like to consider your response to these concerns before we reach a final decision on publication.

We therefore invite you to revise your manuscript to address these concerns. It would be important to carefully address all points raised with additional benchmarking studies, comparison to alternative approaches, inclusion of additional details and expanded discussion.

Link Redacted

We hope to receive your revised paper within 8 weeks. If you cannot send it within this time, please let us know. In this event, we will still be happy to reconsider your paper at a later date so long as nothing similar has been accepted for publication at Nature Methods or published elsewhere.

OPEN SCIENCE REQUIREMENTS

REPORTING SUMMARY AND EDITORIAL POLICY CHECKLISTS

IMAGE INTEGRITY

EXTENDED DATA FIGURES

DATA AVAILABILITY

All novel DNA and RNA sequencing data, protein sequences, genetic polymorphisms, linked genotype and phenotype data, gene expression data, macromolecular structures, and proteomics data must be deposited in a publicly accessible database, and accession codes and associated hyperlinks must be provided in the "Data Availability" section.

Please include a "Data availability" subsection in the Online Methods. This section should inform readers about the availability of the data used to support the conclusions of your study, including accession codes to public repositories, references to source data that may be published alongside the paper, unique identifiers such as URLs to data repository entries, or data set DOIs, and any other statement about data availability. At a minimum, you should include the following statement: "The data that

support the findings of this study are available from the corresponding author upon request”, describing which data is available upon request and mentioning any restrictions on availability. If DOIs are provided, please include these in the Reference list (authors, title, publisher (repository name), identifier, year). For more guidance on how to write this section please see: <http://www.nature.com/authors/policies/data/data-availability-statements-data-citations.pdf>

CODE AVAILABILITY

Please include a “Code Availability” subsection in the Online Methods which details how your custom code is made available. Only in rare cases (where code is not central to the main conclusions of the paper) is the statement “available upon request” allowed (and reasons should be specified).

MATERIALS AVAILABILITY

SUPPLEMENTARY PROTOCOL

To help facilitate reproducibility and uptake of your method, we ask you to prepare a step-by-step Supplementary Protocol for the method described in this paper. We [encourage authors to share their step-by-step experimental protocols](https://www.nature.com/nature-research/editorial-policies/reporting-standards#protocols) on a protocol sharing platform of their choice and report the protocol DOI in the reference list. Nature Portfolio's protocols.io is a free-to-use and open resource for protocols; protocols deposited onto protocols.io are citable and can be linked from the published article. More details can found at [protocols.io](https://www.protocols.io/help/publish-articles).

ORCID

Nature Methods is committed to improving transparency in authorship. As part of our efforts in this direction, we are now requesting that all authors identified as ‘corresponding author’ on published papers create and link their Open Researcher and Contributor Identifier (ORCID) with their account on the Manuscript Tracking System (MTS), prior to acceptance. This applies to primary research papers only. ORCID helps the scientific community achieve unambiguous attribution of all scholarly contributions. You can create and link your ORCID from the home page of the MTS by clicking on ‘Modify my Springer Nature account’. For more information please visit www.springernature.com/orcid.

Sincerely,
Arunima

Arunima Singh, Ph.D.
Senior Editor
Nature Methods

Reviewers' Comments:

Reviewer #1 (Remarks to the Author):

This paper describes a method, FlexConsensus, that is designed to solve an open problem for cryo-EM heterogeneity methods: bringing latent particle embeddings from different methods (or runs of the same method) into register so that they can be compared. The proposed method involves training an autoencoder neural network to map multiple input latent embeddings into a shared “consensus” space, and also decode back to each of the input spaces. This mapping allows for comparison of the input embeddings. Notably, the method does not directly look at particle images or volumes. The method is tested on synthetic and experimental single particle cryo-EM data. The paper suggests that the new method can be used to measure the

reproducibility of the input heterogeneity embeddings and also to filter particle images based on these reproducibility scores.

The problem that the method attempts to solve is important for the field, because at the moment there are no established methods for comparing embeddings; in fact there are no established methods even for validating individual embeddings. To be useful as a practical method, a solution to this problem would need to be clearly interpretable by users and be effective with a variety of input embeddings from different methods.

Overall, while I believe that the method developed in this work is sound and might be useful in practice, I have a few relatively major concerns with the paper, especially in regards to the synthetic and experimental validation of the method. It may be possible for the authors to adjust their approach to account for these concerns, but without substantial changes in the experiments, I don't think the paper gives readers an appropriate level of confidence in the method. I am listing the major points below, and I also have some minor comments, at the end.

Major comments:

1) Dimensionality of the input and consensus embeddings

a) The authors do not mention the fact that, along with the input dimensionality, the "intrinsic" dimensionality of the heterogenous manifold of the dataset is crucial; if FlexConsensus is trained with a smaller number of dimensions in the consensus space than the intrinsic dimensionality, it will be forced to collapse/lose info from one or more input embedding. Instead the authors just use a default of 3 for the consensus space, and mention it can be changed. This is not appropriate in general, and not even in the experiments the authors do where the input dimensions have at least 8 dimensions, and there's no indication that the intrinsic dimensionality is less than 8 (in the experimental datasets). The method should at least use the minimum of the input embeddings as the consensus embedding dimension, and the importance of this choice should be made very clear to the reader. L87 mentions that the dimensionality mismatch between input spaces is important, but there's no discussion of how to actually deal with it (other than simply picking a dimensionality arbitrarily for the consensus space)

2) The synthetic tests in Figure 1 do not provide a lot of evidence towards validation of the method

a) Firstly, by construction, the intrinsic dimension of the generated curves (Equation 1) is only 1. This fact is not mentioned by the authors, but it means that the generated data are a very poor proxy for real cryo-EM embeddings; almost any real heterogeneous dataset will have more than 1 intrinsic dimension. Having only 1 intrinsic dimension also makes the synthetic dataset trivial to learn mappings for, and makes the synthetic test cases not very challenging.

b) Furthermore, in the first synthetic test, the generated points in the two input spaces are generated from equation 1, and therefore are exactly identical in the first 8 dimensions. This means that the optimal mapping into the 3 dimensional consensus space is just to take the first three dimensions of each input embedding, unchanged. Again, this makes the test completely trivial and does not give any evidence towards validating the FlexConsensus method or training algorithm.

c) In fact, in Figure 1 it's clear that the network did not learn to bring the two input embeddings perfectly into register, even though the task is trivial - this is concerning because on such a simple synthetic example (with no noise and identity as the correct solution) if the network and training setup is correct, should we not expect it to get zero error in the consensus space?

d) The second synthetic test is a little bit more challenging but only by a shift and rotation both of which are operations that even a single layer MLP can trivially learn. The synthetic tests, to be convincing, should at least contain a non-linear transformation, forcing the FlexConsensus encoders to learn a non-linear mapping.

e) The second part of the second synthetic test shows that FlexConsensus can't map the very noisy points to each other - but there's nothing in the results shown that helps the reader to believe this should be taken as evidence that FlexConsensus is detecting the noise, rather than simply being poorly optimized or not having a sufficiently powerful network architecture. If a synthetic dataset is being produced to check that FlexConsensus can detect regions of input embeddings that don't have shared structure, then a much better synthetic input could be generated, that e.g. has some continuously changing level of correlation, so that in the FlexConsensus results the reader can see that the method is able to measure the "ground-truth" level of congruency between the input spaces.

3) The idea that FlexConsensus measures reproducibility or accuracy of the input embeddings in a meaningful is not sufficiently evidenced by the results

a) In each result, the comparisons between input embeddings are between only 2 or 3 input embeddings, which does not seem like a sufficient number to draw strong conclusions.

b) Furthermore, there are many possible reasons for FlexConsensus to not find a good match between the input embeddings, and these reasons are not explored in the paper; e.g., it's possible that FlexConsensus is just poorly trained or was not optimized to convergence, or is in a local optimum. These would all lead to measuring substantial error between embeddings in the consensus space.

c) Even if FlexConsensus were completely perfectly trained, the "reproducibility" that the authors describe can be interpreted in many different ways; e.g. in the Ribosome case it was interpreted as being caused by one of the heterogeneity methods (Zernike3D) not being able to model compositional heterogeneity. Then in the spike protein case, it was interpreted as meaning "bad particle images or random estimation errors". Given this, it doesn't make sense to just filter out particles based on "reproducibility" as the authors suggest; in real-life cases the particles with high error under FlexConsensus may be the most important ones. L429 states that filtering yields an embedding that is "more reliable, significant, and accurate" - but there is no evidence to suggest that is true, and as mentioned it is very difficult for a user to know what filtering is doing or whether it is a good idea.

4) L54: It would be important for the introduction to explain that there are not currently any well established or widely used validation techniques for heterogeneity.

5) L67: The idea that FlexConsensus could be used to measure variability between runs is mentioned here and elsewhere. But I don't see any mention of the fact that FlexConsensus itself is a stochastic method and depends on a training procedure that

can yield a different mapping each time it is run; the authors do not claim (and I don't think it would be true) that FlexConsensus will find the globally optimal mapping between the input embeddings. Is it not important that the variability between FlexConsensus runs (with the same inputs) be taken into account, or at least measured in the experiments in the paper?

6) L138: the authors make use of the Earth Mover's Distance (EMD) as a measure of accuracy of the FlexConsensus mappings. It is mentioned that this is appropriate because the input spaces can have different dimensions. However, it is not very clear from the paper why EMD is appropriate in this setting (as compared to other metrics) or exactly how the EMD is computed when the dimensionality of the inputs differ.

7) L172: the authors mention that the histograms in Figure 1 have a "small" error but this is not well defined. In fact it's clear from the method that the values of error are arbitrary, based only on the scale of the input embeddings, and each input embedding can be different. If users are meant to be able to use the FlexConsensus method easily, there should be some way to tell if a certain value of error is "small".

8) L248: the authors mention that the EMD in Figure 2b is "very large" but it is only 0.09, whereas in Figure 2a it is 0.02 - this does not seem like a substantial difference. Again if this method is to be used in the field, these values need some way to be interpretable that is clear and simple.

9) L338: The authors mention that the figures only show a 2D projection of the 3D consensus space. Already, (as explained in point 1 above) it is not clear why the dimensionality of the consensus space should be 3, when the intrinsic dimensionality of the two input embeddings is not known. However, additionally problematic is that it is not made clear to the reader when it is okay to just look at a 2D projection of the consensus space, or when the 3D version (which the authors mention, but don't have a figure for) is necessary. This is important because the purpose of FlexConsensus is to provide a comparison and validation tool for users, so these interpretation questions need to be clear.

Minor comments:

1) L12: experimenting should be experiencing

2) L80: undefined reference

3) L307: The authors use the term "contour" and "isolines" which are somewhat confusing because in the figures, it seems that the aspect they are referring to is really the color values in the 2D histogram (i.e. the heatmap). Also the figures have dotted lines, which at first read seem like the "contours" the authors are referring to. I think a better word could be chosen, for clarity.

Reviewer #2 (Remarks to the Author):

A. Summary of the key results

Several deep-learning based algorithms of analyzing structural heterogeneity from the cryo-EM images have been developed over the past several years. Depending on the complexity of heterogeneity (compositional or conformational heterogeneity) in the cryo-EM data, the conformational space estimations and user's interpretations of them may vary a lot. An algorithm for performing sanity checks is needed in the field of cryo-EM data processing to ensure the correct interpretation of the heterogeneous data. In this study, D. Herreros et al developed an algorithm FlexConsensus to evaluate the agreement among the conformational landscapes generated by different existing methods. The authors built a multi-autoencoder architecture to place different conformational landscapes in a shared consensus space to compare the commonalities and differences among them. The authors further developed several error metrics to evaluate the reliability of the different estimations, allowing the extraction of subsets based on the consensus for further analysis.

B. Originality and significance: if not novel, please include reference

FlexConsensus is the first algorithm, as far as I know, to compare the conformational landscape estimations of cryo-EM images from different existing methods. It offers a valuable metric to assess the heterogeneity estimations from different algorithms and would potentially benefit emerging and ongoing research focused on characterizing dynamics of macromolecules using cryo-EM.

C. Data & methodology: validity of approach, quality of data, quality of presentation

The authors use both the synthetic data (noise or noiseless) and real experimental data to evaluate the performance of FlexConsensus.

For the synthetic dataset, clarification is needed in the description of the data preparation: In generating the simulated data for Figure 2 (line 191 to 206), please clarify for each case (8D and 39 D), which duplicate (translated/ rotated?) has the noise added. In the Figure 2, please describe such modifications to the input spaces by adding figure legends or in the figure caption. In preparing the input spaces from the experimental data, please clarify whether further pose refinement of the particles is performed by HetSIREN? If so, please comment whether this further pose refinement may contribute to the disagreement in input spaces from cryoDRGN and HetSIREN in Figure 5.

D. Appropriate use of statistics and treatment of uncertainties

In Figure 6, please explain the large difference in the representation errors between the two subfigures. Figure 7, please label the Y axis of the permutation distribution plot.

E. Conclusions: robustness, validity, reliability

Please include the cost of the training in terms of the computing resources and time in the methods part.

F. Suggested improvements: experiments, data for possible revision

In this study, the authors demonstrate that FlexConsensus could access the general agreement between input spaces, allowing isolation of the outlier particles not captured by certain method e.g. deformation field-based methods to identify the compositional difference, or those noisy particles in the periphery of the shared consensus landscape as shown in Figures 3 and 5. However, another significant potential of this algorithm would be to explore the disagreement between the input spaces in the central cloud of the consensus space. For example, in the Figure 7, a large majority of particles are not with the significant consensus between two algorithms. A further analysis of such difference would shed light on the reliability of current available methods in identifying continuous heterogeneity. The authors may discuss this part in the manuscript.

G. References: appropriate credit to previous work?

Yes.

H. Clarity and context: lucidity of abstract/summary, appropriateness of abstract, introduction and conclusions

The manuscript is well structured and written. A few typos and errors need to be corrected.

Figure 3b caption: For Zernike3D, the highlighted regions correspond to the region where the biggest conformational variability change has been detected by HetSIREN. Do the authors mean compositional variability?

Typos in Line 314 and 449.

Reviewer #3 (Remarks to the Author):

Summary of the paper

This manuscript introduces FlexConsensus, a novel multi-autoencoder architecture designed to merge conformational landscapes from different heterogeneity analysis algorithms in cryo-electron microscopy (cryo-EM). The authors demonstrated that FlexConsensus can:

1. Create a unified consensus space from disparate conformational landscapes with different dimensionalities.
2. Allow for the identification and filtering of particles based on their stability across different algorithms.
3. Enable conversion between different conformational landscape representations.

The methodology is validated using both synthetic datasets with controlled heterogeneity and real experimental datasets (EMPIAR 10028 and SARS-CoV-2 D614G spike). Overall, FlexConsensus represents a significant contribution to the field of cryo-EM heterogeneity analysis by providing a framework to compare and integrate results from different algorithms.

Suggested improvements

Consideration of related approaches

The manuscript would benefit from a brief discussion of related methods that address different aspects of heterogeneity analysis:

1. "Integrating molecular models into CryoEM heterogeneity analysis using deep neural networks" [1] incorporates prior structural knowledge from molecular modeling. While FlexConsensus focuses on comparing methods, incorporating structural priors could further enhance its biological interpretability.
2. "OPUS-DSD: deep structural disentanglement for cryo-EM" [2] takes a different approach by disentangling complex structural variations into interpretable components. FlexConsensus might benefit from exploring whether consensus spaces could similarly be disentangled into biologically meaningful dimensions.
3. "Cryo-EM heterogeneity analysis using regularized covariance estimation" [3] employs statistical frameworks for heterogeneity analysis. FlexConsensus could potentially incorporate similar statistical measures to provide confidence metrics for its consensus representations.

A high-level discussion of how FlexConsensus relates to these alternative approaches would provide valuable context for readers and highlight the unique contributions of this method within the broader landscape of heterogeneity analysis tools.

Algorithm details and estimation

1. The implementation details of the multi-autoencoder architecture could be more thoroughly described. Specific network architecture choices (number of layers, neurons per layer, activation functions) and hyperparameters (learning rate, batch size, regularization parameters, optimizer choice) should be included.
2. The manuscript would be significantly strengthened by implementing simpler baseline methods for comparison. A systematic benchmarking against classical dimensionality reduction techniques such as t-SNE and simple distance-based clustering of conformations.

Specific experimental additions

1. Resolution assessment: Analyze whether particles identified as having high consensus error affect the resolution of reconstructed volumes. Apply different consensus error thresholds (removing 5%, 10%, 15% of high-error particles), perform 3D reconstruction, calculate FSC curves, and report resolution changes at different filtering levels.

2. I recommend comparing FlexConsensus with standardized benchmark datasets from CryoBench [4] to better evaluate performance on well-characterized heterogeneity challenges:

IgG-RL dataset: This dataset models random configurations of an antibody linker peptide, creating complex, non-linear conformational variability. Testing FlexConsensus on this dataset would validate its ability to handle disordered regions and multiple conformational degrees of freedom, particularly demonstrating whether the consensus space can effectively capture the relationship between center-of-mass distance and dihedral angle parameters that characterize the ground truth. The authors should specifically analyze whether the consensus space dimensions correlate with the physical parameters of the linker motion, and whether particles with high consensus error correspond to conformational extremes or artifacts.

Spike-MD dataset: Derived from molecular dynamics simulations of SARS-CoV-2 spike protein (46,789 unique conformations), this dataset would challenge FlexConsensus with subtle, continuous conformational changes across multiple scales. This test would demonstrate whether the consensus metrics can identify meaningful biological transitions such as the receptor binding domain opening, and whether the consensus space dimensions correlate with the established collective variables (CV1/CV2) that parameterize the conformational landscape. The authors should map the consensus space onto known energy landscapes from the MD simulation to assess whether the method captures energetically relevant states and transitions. For more rigorous validation, data from CryoBench could serve as a reference for performance assessment. The authors could select relevant datasets for further validation and quantify performance using established metrics like neighborhood similarity and information imbalance.

Figure improvements

1. The uploaded figures are not vector graphics, causing pixelation when zoomed.
2. Add visualization comparisons of the consensus space against simpler approaches such as t-SNE and simple distance-based clustering of conformations.

User accessibility

More discussion on how structural biologists can practically incorporate this tool into their workflow would enhance the manuscript's impact.

[1] Chen M, Toader B, Lederman R. Integrating molecular models into CryoEM heterogeneity analysis using scalable high-resolution deep Gaussian mixture models[J]. Journal of molecular biology, 2023, 435(9): 168014.

[2] Luo Z, Ni F, Wang Q, et al. OPUS-DSD: deep structural disentanglement for cryo-EM single-particle analysis[J]. Nature Methods, 2023, 20(11): 1729-1738.

[3] Gilles M A, Singer A. Cryo-EM heterogeneity analysis using regularized covariance estimation and kernel regression[J]. Proceedings of the National Academy of Sciences, 2025, 122(9): e2419140122.

[4] Jeon M, Raghu R, Astore M, et al. CryoBench: Diverse and challenging datasets for the heterogeneity problem in cryo-EM[J]. Advances in Neural Information Processing Systems, 2024, 37: 89468-89512.

Reviewer #3 (Remarks on code availability):

Code is available through Scipion 3.0 via scipion-em-flexutils plugin with specific FlexConsensus protocols.

Version 1:

Decision Letter:

Our ref: NMETH-A59327A

5th Jun 2025

Dear Dr. Herreros,

Thank you for submitting your revised manuscript "Merging conformational landscapes in a single consensus space with FlexConsensus algorithm" (NMETH-A59327A). It has now been seen by the original referees and their comments are below. The reviewers find that the paper has improved in revision, and therefore we'll be happy in principle to publish it in Nature Methods, pending minor revisions to satisfy the referees' final requests and to comply with our editorial and formatting guidelines.

TRANSPARENT PEER REVIEW

Please note: we allow redactions to authors' rebuttal and reviewer comments in the interest of confidentiality. If you are

concerned about the release of confidential data, please let us know specifically what information you would like to have removed. Please note that we cannot incorporate redactions for any other reasons. Reviewer names will be published in the peer review files if the reviewer signed the comments to authors, or if reviewers explicitly agree to release their name. For more information, please refer to our [FAQ page](https://www.nature.com/documents/nr-transparent-peer-review.pdf).

ORCID

Sincerely,
Arunima

Arunima Singh, Ph.D.
Senior Editor
Nature Methods

Reviewer #1 (Remarks to the Author):

Response to the author's rebuttal:

- 1)
 - a)

While I feel that the discussion of intrinsic dimensionality of the latent spaces is still very important for a method like FlexConsensus, the change of the default to use the min of input dimensions is a good step. There may be a typo, but the rebuttal document says that the default is "automatically set the number of dimensions to the minimum of all the input embeddings", but new manuscript text still says that the default is three. This should be corrected. Given that the purpose of FlexConsensus is not just to approximately match the input latent embeddings to each other, but to measure and compare them, I think it's important that another sentence be added to the manuscript to the effect that "Setting a consensus space dimension smaller than the intrinsic dimensionality of the input latent dimensions means that some information from the inputs will potentially be ignored when mapping to the consensus space."
- 2)

In the rebuttal text, the authors explain that the methods that produced the input spaces may have failed to capture the variation of the particle; but that is not what this point was about. The point is that the synthetic test cases in the original draft were really not complex or difficult enough to provide evidence that the method works. The new synthetic example is better than the original examples - the intrinsic dimensionality of one of the inputs is at least greater than 1, and the two inputs do not have any identical dimensions. The mapping to consensus space is clearly non-linear. So it is a much better test in my opinion and I think helps improve the paper. This new synthetic test in combination with the examples on real data I think are now sufficient for readers to be confident in the method.
- 3)
 - a)

Thank you, I can agree with the author's argument.
 - b) c)

The new text and experiment is useful. Showing additional analysis using the reproducibility as defined as consensus error helps to confirm that in practice it measures something that is meaningful for the dataset. The additional comparison with 3D classification is also helpful to make concrete the meaning of error in the consensus space. The added text also helps the reader to understand the difference between "bad" particles and estimation errors in the input heterogeneity methods.
- 4) Thank you for making the suggested change.
- 5) Thank you, the addition to the supplementary materials helps clarify.
- 6) Thank you, the additional text helps clarify.
- 7) Thank you, the new method for normalizing makes comparisons of size more clear.
- 8) Thank you for the change.
- 9) Thank you, the new section about the workflow is an important and useful addition.

Minor comments all addressed.

Based on the changes, I would be happy to see the paper proceed.

Reviewer #2 (Remarks to the Author):

The authors have addressed the previous concerns raised.

A few more questions need to be clarified in the manuscript.

1. Supplementary Figure 6 shows that the particles with higher consensus errors are not necessarily "bad" particles; rather, they may represent images that are incorrectly assigned in the conformational landscape by individual heterogeneity methods. This raises the question: if these particles are more accurately estimated in a high-dimensional conformational space, could they be mistakenly assigned when projected into a lower-dimensional consensus space? In other words, how much does the choice of consensus space dimensionality influence the distribution of consensus errors? The author may need to clarify this in the manuscript.

2. In the revised manuscript, lines 57–65, the authors discuss the current limitations of validation of heterogeneity methods. Later, in lines 938–942, they introduce the 3D space annotator, which is used to generate real-time conformations from any region within the consensus space. However, clarification is needed regarding whether the 3D space annotator outputs a single volume per particle with high confidence. Specifically, is the interpretability of the output still constrained by the number of particles surrounding the center of a given class?

Reviewer #3 (Remarks to the Author):

A. Summary of the key results

The authors have successfully addressed all my concerns from the first review. The revised manuscript now presents FlexConsensus as a comprehensive tool for merging conformational landscapes from different cryo-EM heterogeneity analysis methods, with robust validation on both synthetic and experimental datasets.

B. Originality and significance

The significance of this work remains high. The additional experiments on CryoBench datasets further strengthen the method's validity and demonstrate its broad applicability to challenging heterogeneity problems in cryo-EM.

C. Data & methodology: validity of approach, quality of data, quality of presentation

- The authors have significantly improved the methodology section with following:
- Detailed network architecture description (3 fully connected layers, 1024 neurons each, ReLU activation)
- Clear hyperparameter specifications (learning rate 1e-5, batch size 1024, Adam optimizer)
- Computational requirements (16 seconds/epoch on RTX Ada 6000 GPU, ~2.6GB VRAM)
- Comprehensive comparison with t-SNE and distance-based clustering (Supplementary Figure 1)

D. Appropriate use of statistics and treatment of uncertainties

The statistical analyses are now robust, particularly the Mantel test for IgG-RL dataset ($p=0.002$) and the correlation analysis with collective variables for the MDSpoke dataset.

E. Conclusions: robustness, validity, reliability

The conclusions are well-supported by the expanded experimental evidence. The finding that high consensus error particles are not "bad" particles but rather incorrectly estimated by the methods is particularly insightful.

F. Suggested improvements

The authors have comprehensively addressed all my previous suggestions:

- 1) Related approaches discussion: Now included in the Introduction with clear differentiation of FlexConsensus's unique positioning
- 2) Algorithm details: Fully specified with architecture, hyperparameters, and computational requirements
- 3) CryoBench validation: Excellent analysis of both IgG-RL and MDSpoke datasets with correlation to ground truth parameters
- 4) Resolution assessment: FSC curves demonstrate minimal impact of filtering on resolution
- 5) Figure quality: Improved to vector graphics with higher resolution
- 6) User accessibility: New Methods section on Scipion integration with tutorial links

G. References: appropriate credit to previous work?

Yes, appropriate references have been added.

H. Clarity and context

The manuscript is now significantly clearer with improved organization and the addition of practical workflow integration details.

Additional Comments

The authors provided thorough and thoughtful responses to my previous review comments. The additional experiments, particularly the CryoBench analyses, provided strong validation of the method. The correlation between consensus space and physical parameters (center-of-mass distance and dihedral angles for IgG-RL, collective variables for MDSpoke) convincingly demonstrates that FlexConsensus captures biologically meaningful information.

The practical improvements (Scipion integration, tutorials, interactive workflow visualization) will greatly enhance the tool's adoption by the cryo-EM community.

Reviewer #3 (Remarks on code availability):

The code is available through the provided GitHub repositories. The addition of tutorials and interactive workflow visualization significantly improves accessibility.

Version 2:

Decision Letter:

21st Aug 2025

Dear David,

I am pleased to inform you that your Article, "Merging conformational landscapes in a single consensus space with FlexConsensus algorithm", has now been accepted for publication in Nature Methods. The received and accepted dates will be January 7, 2025 and August 21, 2025. This note is intended to let you know what to expect from us over the next month or so, and to let you know where to address any further questions.

Over the next few weeks, your paper will be copyedited to ensure that it conforms to Nature Methods style. Once your paper is typeset, you will receive an email with a link to choose the appropriate publishing options for your paper and our Author Services team will be in touch regarding any additional information that may be required. It is extremely important that you let us know now whether you will be difficult to contact over the next month. If this is the case, we ask that you send us the contact information (email, phone and fax) of someone who will be able to check the proofs and deal with any last-minute problems.

Authors may need to take specific actions to achieve compliance with funder and institutional open access mandates.

If your research is supported by a funder that requires immediate open access (e.g. according to [Plan S principles](https://www.springernature.com/gp/open-science/plan-s-compliance) or the [NIH public access policy](https://www.springernature.com/gp/open-science/us-federal-agency-compliance)) then you should select the gold OA route, and we will direct you to the compliant route where possible. Because authors warrant under our subscription licensing terms that they haven't committed to licensing any version of their article under a licence inconsistent with the terms of our agreement – including the applicable embargo period – publication under the subscription model isn't suitable for authors whose funders require no embargo.

If you are active on Twitter/X or Bluesky, please e-mail me your and your coauthors' handles so that we may tag you when the paper is published.

Please note that you and any of your coauthors will be able to order reprints and single copies of the issue containing your article through Nature Portfolio's reprint website, which is located at <http://www.nature.com/reprints/author-reprints.html>. If there

are any questions about reprints please send an email to author-reprints@nature.com and someone will assist you.

Best regards,
Arunima

Arunima Singh, Ph.D.
Senior Editor
Nature Methods

** Visit the Springer Nature Editorial and Publishing website at http://editorial-jobs.springernature.com?utm_source=ejP_NMeth_email&utm_medium=ejP_NMeth_email&utm_campaign=ejp_Nmeth > www.springernature.com/editorial-and-publishing-jobs for more information about our career opportunities. If you have any questions please click [here](mailto:editorial.publishing.jobs@springernature.com) . **

RESPONSE LETTER

In the following document, we include our detailed answers to the reviewer's comments, including a description of the modifications in the manuscript to simplify their tracking in the new version. The letter is formatted so the reviewer's comments are highlighted in blue, followed by the answers in dark green. For the completeness of the response letter, we also include the original text added in the reviewed version of the manuscript in purple at the end of every answer.

Reviewer #1

Remarks to the Author

This paper describes a method, FlexConsensus, that is designed to solve an open problem for cryo-EM heterogeneity methods: bringing latent particle embeddings from different methods (or runs of the same method) into register so that they can be compared. The proposed method involves training an autoencoder neural network to map multiple input latent embeddings into a shared "consensus" space, and also decode back to each of the input spaces. This mapping allows for comparison of the input embeddings. Notably, the method does not directly look at particle images or volumes. The method is tested on synthetic and experimental single particle cryo-EM data. The paper suggests that the new method can be used to measure the reproducibility of the input heterogeneity embeddings and also to filter particle images based on these reproducibility scores.

The problem that the method attempts to solve is important for the field, because at the moment there are no established methods for comparing embeddings; in fact there are no established methods even for validating individual embeddings. To be useful as a practical method, a solution to this problem would need to be clearly interpretable by users and be effective with a variety of input embeddings from different methods.

Overall, while I believe that the method developed in this work is sound and might be useful in practice, I have a few relatively major concerns with the paper, especially in regards to the synthetic and experimental validation of the method. It may be possible for the authors to adjust their approach to account for these concerns, but without substantial changes in the experiments, I don't think the paper gives readers an appropriate level of confidence in the

method. I am listing the major points below, and I also have some minor comments, at the end.

Major comments

1) Dimensionality of the input and consensus embeddings

a) The authors do not mention the fact that, along with the input dimensionality, the “intrinsic” dimensionality of the heterogenous manifold of the dataset is crucial; if FlexConsensus is trained with a smaller number of dimensions in the consensus space than the intrinsic dimensionality, it will be forced to collapse/lose info from one or more input embedding. Instead the authors just use a default of 3 for the consensus space, and mention it can be changed. This is not appropriate in general, and not even in the experiments the authors do where the input dimensions have at least 8 dimensions, and there’s no indication that the intrinsic dimensionality is less than 8 (in the experimental datasets). The method should at least use the minimum of the input embeddings as the consensus embedding dimension, and the importance of this choice should be made very clear to the reader. L87 mentions that the dimensionality mismatch between input spaces is important, but there’s no discussion of how to actually deal with it (other than simply picking a dimensionality arbitrarily for the consensus space)

As the reviewer indicates, choosing the appropriate dimension for network latent spaces is challenging, as there is usually no a priori information on their intrinsic dimensionality.

Our decision to use three as the default dimension is based on representation purposes, thereby avoiding the need to reduce the dimensionality of the consensus space further when presented to the user. However, we agree that this decision may result in a loss of accuracy due to the collapse of information into a small number of dimensions.

Therefore, as suggested, we have modified the protocol in Scipion to automatically set the number of dimensions to the minimum of all the input embeddings. This option allows users to choose the consensus space dimension sensibly if they prefer not to set it manually.

We have also discussed these changes in the manuscript. We include below the newly added text for completeness:

We note that FlexConsensus learns a consensus space with three dimensions by default. However, the latent space dimensionality is exposed as a customizable parameter in the corresponding Scipion

(14) protocol form. All the consensus landscapes presented in the following sections were defined to have three dimensions as set by default.

Regarding L87, FlexConsensus can address the dimensionality problem thanks to its multi-encoder architecture. This architecture allows us to learn a set of features for each input space based solely on its characteristics. These features are then fed to a dense layer, which maps them to the consensus space.

2) The synthetic tests in Figure 1 do not provide a lot of evidence towards validation of the method

a) Firstly, by construction, the intrinsic dimension of the generated curves (Equation 1) is only 1. This fact is not mentioned by the authors, but it means that the generated data are a very poor proxy for real cryo-EM embeddings; almost any real heterogeneous dataset will have more than 1 intrinsic dimension. Having only 1 intrinsic dimension also makes the synthetic dataset trivial to learn mappings for, and makes the synthetic test cases not very challenging.

b) Furthermore, in the first synthetic test, the generated points in the two input spaces are generated from equation 1, and therefore are exactly identical in the first 8 dimensions. This means that the optimal mapping into the 3 dimensional consensus space is just to take the first three dimensions of each input embedding, unchanged. Again, this makes the test completely trivial and does not give any evidence towards validating the FlexConsensus method or training algorithm.

c) In fact, in Figure 1 it's clear that the network did not learn to bring the two input embeddings perfectly into register, even though the task is trivial - this is concerning because on such a simple synthetic example (with no noise and identity as the correct solution) if the network and training setup is correct, should we not expect it to get zero error in the consensus space?

d) The second synthetic test is a little bit more challenging but only by a shift and rotation both of which are operations that even a single layer MLP can trivially learn. The synthetic tests, to be convincing, should at least contain a non-linear transformation, forcing the FlexConsensus encoders to learn a non-linear mapping.

e) The second part of the second synthetic test shows that FlexConsensus can't map the very noisy points to each other - but there's nothing in the results shown that helps the reader to believe this should be taken as evidence that FlexConsensus is detecting the noise, rather than

simply being poorly optimized or not having a sufficiently powerful network architecture. If a synthetic dataset is being produced to check that FlexConsensus can detect regions of input embeddings that don't have shared structure, then a much better synthetic input could be generated, that e.g. has some continuously changing level of correlation, so that in the FlexConsensus results the reader can see that the method is able to measure the "ground-truth" level of congruency between the input spaces.

This is a very important point, and we thank the reviewer for bringing this issue to our attention. We have a complete answer for it.

In general, FlexConsensus sees the original intrinsic space through the eyes of the latent spaces of the input methods. If these methods fail to capture the original variations fully, then FlexConsensus cannot match them. This issue is why, in Fig. 1, the two trajectories do not precisely match. On top of this, we have the limitations imposed by a particular network architecture and optimization procedure. Neural networks have been proven to be universal function approximators under relatively mild conditions. However, this result applies to networks of potentially infinite size and assumes that the optimization procedure can reliably find a global minimum. In practice, neural networks are finite, and training algorithms can become trapped in local minima or saddle points. While this limitation is inherent to any numerical optimization method, we have not found it to pose significant issues in our experiments.

Based on the reviewer's suggestions, we have included a new synthetic dataset that is more challenging according to the indications stated in the previous comments. This new dataset will focus on generating input spaces with different intrinsic dimensions, distinct by construction, and with more apparent corruption of the input spaces to better assess FlexConsensus's capabilities.

We have defined a new set of two input spaces to overcome all the previous issues. The first one is based on a helical curve, which has an intrinsic dimension equal to one:

$$f(t) = (\cos(t), \sin(t), t, \sin(4t), \sin(5t), \dots)$$

Where t takes values from $[0, 2c\pi]$, the constant c allows the control of the number of turns of the helical curve. In addition, half of the turns of the curve are further modified to introduce a non-linear stretching that depends on the distance to the center of the helix, followed by a

corruption of this stretched region with Gaussian noise whose standard deviation also depends on the previous distance:

$$f(t) = \begin{cases} (\sin(t), \cos(t), t, \sin(4t), \dots), & t \leq c\pi \\ (\sin(t) + \sigma(r), \cos(t) + \sigma(r), t + \sigma(r), \sin(4t) \cos(5t) + \sigma(r), \dots), & t > c\pi \end{cases}$$

The second space is defined as a helix-shaped surface with intrinsic dimensionality equal to two:

$$f(t) = (a \cos(t), a \sin(t), t, \sin(4t) \cos(5t), \sin(5t) \cos(6t), \dots)$$

Where t takes values from $[0, 2c\pi]$ and a is a random number drawn from a uniform distribution. The two spaces simulated had eight turns each, with a total of 5000 points and dimensions 20 and 40, respectively. The results obtained from this experiment are summarized in the Figure below:

Figure 3: FlexConsensus analysis resulting from comparing two synthetic spaces (in blue and red) with different intrinsic dimensions based on a helix curve and a helix-shaped surface. In addition, half of the turns in the helix-shaped space are stretched and corrupted with noise depending on the distance to the center of the helix. The resulting consensus space (of dimension 2) shows the capacity of the method to identify the common and uncommon regions present in the input spaces. To the right of the Figure, we present the two isolated consensus spaces obtained from the inputs, with those points estimated to have the largest consensus errors highlighted in yellow.

At the beginning of the Figure, we present an extract from the original spaces showing their first and third dimensions. The first three dimensions were aligned and scaled for representation purposes to simplify their comparison, although these transformations were not applied to the spaces input to FlexConsensus.

The resulting consensus space shows that the method has correctly identified the input spaces' similarities and differences. The regions with the highest consensus error are located in the deformed and noisy regions. Interestingly, the method can still detect that the deformation does not make the spaces intrinsically different.

We have included the previous results and discussion as a new section in the manuscript. We include this new section for completeness:

Consensus results on synthetic spaces with different intrinsic dimensions

The dataset previously analyzed represents a simplified illustrative case in which the intrinsic dimensionality of the spaces was very low (just 1), even if the samples were forced to live in spaces with different dimensions. Consequently, although the previous setup helped conceptualize the method, it fails to capture the complex relationships that may be present in real scenarios where the intrinsic dimensionality of each input space is unknown and might be different for each method.

Therefore, we conducted an additional experiment on a new synthetic dataset that, by construction, had input spaces of different intrinsic dimensionality. In addition, we imposed several non-linear constraints to better evaluate the performance of FlexConsensus in a more realistic but still controlled scenario.

The new dataset comprises two different spaces, the first based on a helix curve:

$$f(t) = (\cos(t), \sin(t), t, \sin(4t), \sin(5t), \dots)$$

Where t takes values in $[0, 2c\pi]$, the constant c allows the control of the total number of turns of the helix. In addition, half of the turns of the previous space were further modified to introduce a non-linear transformation that helps evaluate whether FlexConsensus can detect this dissimilarity. The proposed non-linear transformation consists of a non-linear deformation that stretches the helix from its middle turn onward, whose strength depends on the distance to the center of the helix. In addition to this non-linear deformation, noise is added to these stretched regions. The noise added follows a normal distribution whose standard deviation depends on the distance to the center of the helix:

$$f(t) = \begin{cases} (\sin(t), \cos(t), t, \sin(4t), \dots), & t \leq c\pi \\ (\sin(t) + \sigma(r), \cos(t) + \sigma(r), t + \sigma(r), \sin(4t) \cos(5t) + \sigma(r), \dots), & t > c\pi \end{cases}$$

Being $\sigma(r)$, the deformation and noise added to the original space depending on the distance to the center of the helix r .

The second space is based on a helix-shaped surface defined by:

$$f(t) = (a \cos(t), a \sin(t), t, \sin(4t) \cos(5t), \sin(5t) \cos(6t), \dots)$$

Where t takes values in $[0, 2c\pi]$, and a is a random number taken from a uniform distribution. Similarly to the case before, the constant c allows the control of the total number of turns of the helix.

As can be seen from the equations of the new spaces, apart from having an intrinsic different dimensionality, they have been constructed without coincident dimensions, trying to make them more challenging. The two spaces generated consist of 5000 points each and have 20 and 40 dimensions, respectively, and eight turns. After analyzing these spaces with FlexConsensus, the results are summarized in Figure 3.

On the left-hand side of the Figure, we present a subregion of the two original spaces around the middle point of each space. To simplify their visualization, the first three dimensions of each space were aligned and scaled so they could be more easily compared. It should be noted that the spaces input to the network were not aligned or scaled by any means.

Regarding the consensus space learned by the network, the central image from Figure 3 shows that FlexConsensus has correctly identified that the initial turns from each space are similar. In contrast, the last turns are assigned a more significant consensus error, as expected. This result shows the ability of the method to identify the similarities and differences between the input spaces, even if they have a different intrinsic number of dimensions. Interestingly, the consensus space also shows that FlexConsensus neglects the non-linear deformation added to the final half of the turns. It correctly identifies that it should "stretch" those turns in the helix-shaped surface to match the helix curve samples. This result confirms that FlexConsensus should not be affected by non-linear modifications of the input spaces as long as the position of the samples in these spaces is consistent.

Complementing the previous results, we present the isolated consensus space for each input on the right-hand side of the Figure, highlighting in yellow those points identified to have a significant consensus error. As expected, the samples identified to have a larger consensus error are located in the half turns that were more different by construction (i.e., the region modified with noise and a non-linear deformation in the helix curve dataset). Therefore, it is possible to conclude that FlexConsensus can correctly estimate which regions in the input spaces are compatible or incompatible.

3) The idea that FlexConsensus measures reproducibility or accuracy of the input embeddings in a meaningful is not sufficiently evidenced by the results

a) In each result, the comparisons between input embeddings are between only 2 or 3 input embeddings, which does not seem like a sufficient number to draw strong conclusions.

We agree with the reviewer that additional estimations on the landscape will enhance the quality of the consensus results, enabling more accurate conclusions and stronger statistical analysis.

Although it would be possible to execute the same method several times, we believe that the most interesting case is comparing different methods to better understand their similarities and differences. We have integrated four methods (Zernike3D, HetSIREN, CryoDRGN, and 3DFlex) in Scipion. We are constantly working on integrating new software, and some methods, such as Dnyamight, Opus-DSD, or RecoVar, will soon be compatible with Scipion and FlexConsensus.

Therefore, we hope that more methods will soon be included in the analysis, allowing users to execute as many estimations as needed to improve the consensus analysis.

Additionally, from a practical perspective, demonstrating that the consensus can be implemented with fewer methods is interesting and probably of direct use on many occasions. In general, users do not execute a wide range of different methods in their projects. Thus, showing the method's performance in these cases will help show how results can be analyzed when the number of executions is larger than one, which is the current practice.

b) Furthermore, there are many possible reasons for FlexConsensus to not find a good match between the input embeddings, and these reasons are not explored in the paper; e.g., it's possible that FlexConsensus is just poorly trained or was not optimized to convergence, or is in a local optimum. These would all lead to measuring substantial error between embeddings in the consensus space.

Since FlexConsensus training is relatively short, we chose a large number of epochs to ensure that the training curves stabilized and the model converged. We include below the training curve for one experiment included in the manuscript as a reference:

Users also have access to these training curves through the Scipion interface. In addition, they can check the overall epoch loss and how the losses of the encoder and each decoder in FlexConsensus evolve with the number of epochs.

Regarding the local optimum issue, we agree that this is an inherent problem in neural networks and optimization tasks, as it is impossible to guarantee that the minimum detected during training is a sufficiently good local optimum.

Still, we hope to reach a relatively stable minimum. We discuss this further to answer the reviewer's comment on the reproducibility of the consensus result. We hope that this discussion is useful in assessing the local minima problem.

c) Even if FlexConsensus were completely perfectly trained, the “reproducibility” that the authors describe can be interpreted in many different ways; e.g. in the Ribosome case it was interpreted as being caused by one of the heterogeneity methods (Zernike3D) not being able to model compositional heterogeneity. Then in the spike protein case, it was interpreted as meaning “bad particle images or random estimation errors”. Given this, it doesn’t make sense to just filter out particles based on “reproducibility” as the authors suggest; in real-life cases the particles with high error under FlexConsensus may be the most important ones. L429 states that filtering yields an embedding that is “more reliable, significant, and accurate” - but there is no evidence to suggest that is true, and as mentioned it is very difficult for a user to know what filtering is doing or whether it is a good idea.

As the reviewer suggests, it is interesting to analyze not only the particles estimated to have a low consensus error but also those regions with a larger discrepancy to understand the source of these estimations better.

The filtering methods proposed in this work can be used for both purposes, as it is possible to recover either the "positive" subset of images (i.e. the subset of images determined to have a

low consensus error) as well as the "negative" subset (i.e., those images that the filtering algorithm would remove). This selection can be made easily using the tools provided inside Scipion, thanks to its subset tools, allowing the registration of all these outputs to analyze them further using standard packages.

For example, following the third reviewer's suggestions, we have included an analysis of the reconstructions obtained from some "negative" subsets obtained by thresholding the consensus error histograms at different thresholds.

We include below the new text for completeness:

Consensus results on the EMPIAR 10028 dataset

To better understand the source of the differences detected by FlexConsensus between the two HetSIREN executions and Zernike3D, an additional experiment was conducted to analyze at volume level the conformations arising from the landscape regions detected to have the largest errors. To that end, a subset of the HetSIREN and Zernike3D landscapes was extracted based on the indices obtained from the images estimated to have the most significant error based on the consensus. Posteriorly, the subset spaces were clustered to get ten different representatives of the conformations, which were used to decode their associated conformations with HetSIREN and Zernike3D. The different conformations obtained are presented in Supplementary Video 3. As can be seen from the comparison of the states recovered by both methods, HetSIREN tends to detect motions with a lower amplitude and a strong compositional component. In comparison, Zernike3D estimates continuous motions with a larger amplitude. This result suggests that the consensus differences are not related to "wrong" images but to estimations differences between the two methods.

Consensus results on SARS-CoV-2 D614G dataset

A question remains on analyzing the characteristics of those images estimated to have a large consensus error. It is important to understand whether they correspond to "bad" particles or, instead, their position in conformational space arises from estimation errors from the heterogeneity methods. Therefore, an additional experiment was conducted to compare the reconstructions obtained from those images estimated to have a consensus error larger than a given threshold. The results obtained in this analysis are summarized in Supplementary Figure 6. As shown in the Figure, and as expected, these particles are distributed differently in the HetSIREN and CryoDRGN consensus spaces. In CryoDRGN, the images are placed close to the two-up region, while HetSIREN tends to distribute them along the one-up and three-down states. In contrast, the volume reconstructed from these images

yields a clear one-up state, indicating that the quality of these images was not intrinsically bad but that, for yet unknown reasons, both HetSIREN and CryoDRGN did not properly position these images in conformational space close to other images in similar states of RBD opening. Although the conceptual distinction between images with large consensus errors being either “intrinsically bad” or “badly estimated by a method” is very important, its practical usage is limited.

Supplementary Figure 6: Reconstruction obtained from SARS-CoV-2 dataset considering only those images estimated to be within the 5% largest consensus error (around 21k particles from the original 440k). The landscapes show the distribution of these images in the consensus space: For CryoDRGN, most of these images are located in the two-up state region, while in HetSIREN, they are distributed along the regions corresponding to the one-up and three-down states. In contrast, the reconstruction calculated directly from these particles corresponds to a clear one-up state, indicating that the two heterogeneity methods wrongly assigned these images to regions of the conformational landscape.

Regarding the claim in L429, we have extended the analysis, including a thorough comparison against a 3D classification consensus workflow, which will be assumed to be our “ground truth”. The results are the following:

- 3D classification (Relion (19)): three down (20%), one up (60%), and two up (20%).
- HetSIREN (FlexConsensus): Three down (22%), one up (50%), and two up (28%).
- CryoDRGN (FlexConsensus): Three down (27%), one up (35%), and two up (38%).
- FlexConsensus filtered landscape: Three down (26%), one up (41%), and two up (31%).

This result suggests that HetSIREN tends to follow a similar trend to the 3D classification, unlike CryoDRGN, which assigns a larger weight to the two-up state. The filtered landscape normalizes the original populations, decreasing the two-up states and yielding a trend that aligns with 3D classification.

Therefore, the previous tests indicate that HetSIREN and CryoDRGN estimations were intrinsically different even in the main cloud, as corroborated by visually inspecting the landscapes. However, by keeping only those images estimated similarly by both methods, FlexConsensus seems to retrieve a more normalized distribution of states closer to the 3D classification results.

We include below the new discussion for completeness:

When comparing the results obtained from the two analyses presented in Supplementary Figure 5 and Figure 8, it is possible to see that both approaches detect that the distribution of states detected by HetSIREN and CryoDRGN was incompatible in an important fraction of the images. These results suggest the importance of validating the landscapes estimated by a given algorithm to detect these inconsistencies.

However, it is interesting to understand whether the filtered FlexConsensus space yields more meaningful landscape populations. To that end, we conducted a 3D classification of the original images into several classes, which were posteriorly merged according to the RBD conformational state they represent. Since both HetSIREN and CryoDRGN suggest a clear tendency towards the presence of three well-differentiated conformational states, the populations of the 3D classification will be considered as the "ground truth" populations as they are obtained from a standard approach in the field.

In the case of the landscapes, Gaussian Mixture-based clustering was used to compute the populations. To that end, three Gaussian components were estimated to cluster the spaces to get the relative populations to compare against the 3D classification ones. It should be noted that the populations were directly measured in the consensus subspaces of HetSIREN and CryoDRGN, which were obtained using FlexConsensus.

The relative populations obtained were:

- 3D classification (Relion (19)): three down (20%), one up (60%), and two up (20%).
- HetSIREN (FlexConsensus): three down (22%), one up (50%), and two up (28%).

- CryoDRGN (FlexConsensus): three down (27%), one up (35%), and two up (38%).
- FlexConsensus filtered landscape: three down (26%), one up (41%), and two up (31%).

By comparing the different populations against the 3D classification, it is possible to see how HetSIREN tends to be more conservative, estimating a set of populations that follow a similar trend to the 3D classification ones (i.e., a larger number of one-up states compared to the other two). In contrast, the trend of CryoDRGN is to give a larger population to the two-up states. Flexconsensus filtered landscape tends to normalize the populations to a value that is in between the two original methods, leading to a new set that resembles more the behavior seen in the classification results.

A question remains on analyzing the characteristics of those images estimated to have a large consensus error. It is important to understand whether they correspond to "bad" particles or, instead, their position in conformational space arises from estimation errors from the heterogeneity methods. Therefore, an additional experiment was conducted to compare the reconstructions obtained from those images estimated to have a consensus error larger than a given threshold. The results obtained in this analysis are summarized in Supplementary Figure 6. As shown in the Figure, and as expected, these particles are distributed differently in the HetSIREN and CryoDRGN consensus spaces. In CryoDRGN, the images are placed close to the two-up region, while HetSIREN tends to distribute them along the one-up and three-down states. In contrast, the volume reconstructed from these images yields a clear one-up state, indicating that the quality of these images was not intrinsically bad but that, for yet unknown reasons, both HetSIREN and CryoDRGN did not properly position these images in conformational space close to other images in similar states of RBD opening. Consequently, although the conceptual distinction between images with large consensus errors being either intrinsically bad or badly estimated by a method is very important, it seems we do not have a clear way to distinguish between these two options.

4) L54: It would be important for the introduction to explain that there are not currently any well established or widely used validation techniques for heterogeneity.

Following the reviewer suggestions, we have further discussed in the introduction the current state of validation techniques for heterogeneity in CryoEM. We include below the new text for completeness:

The variety of methods, approaches, and implementations allows one to explore the structural variability of any given dataset through a systematic approach. However, the large pool of existing algorithms also presents the challenge of comparing different results. Currently, there is a lack of validation tools in the CryoEM heterogeneity field. Indeed, the most widely used validation approach

is to reconstruct states from a small subset of images surrounding a given conformation in the landscape and to compare them with the conformations estimated by a given heterogeneity analysis method. However, this approximation is limited by the minimum number of particles needed to reconstruct a volume with enough resolution to detect a given conformational state. If the number of images required is too large, contamination from other structural changes in the reconstruction will prevent a complete validation of the conformational space. Furthermore, the accuracy of current heterogeneity analysis methods remains a subject of ongoing investigation. Therefore, it is crucial to extract consensus solutions from different approaches.

5) L67: The idea that FlexConsensus could be used to measure variability between runs is mentioned here and elsewhere. But I don't see any mention of the fact that FlexConsensus itself is a stochastic method and depends on a training procedure that can yield a different mapping each time it is run; the authors do not claim (and I don't think it would be true) that FlexConsensus will find the globally optimal mapping between the input embeddings. Is it not important that the variability between FlexConsensus runs (with the same inputs) be taken into account, or at least measured in the experiments in the paper?

We agree with the reviewer that neural networks are stochastic, yielding a different result every time they are executed. Additionally, the optimization procedure typically does not reach the global minimum.

Therefore, FlexConsensus estimations are expected to differ when training different networks. However, we would like these differences to be small so that all the estimations resemble a consensus space with similar characteristics.

To that end, we have included in the Supplementary Information a more thorough discussion on the resemblance of the consensus conformational spaces when executed several times, considering that the distribution of states of all the landscapes should be similar. The test relied on training five independent FlexConsensus networks with the helix-based synthetic dataset introduced previously in this response letter. The learned consensus spaces are then compared using the information imbalance measure to determine whether they contain similar information. The main results from this analysis are summarized in Supplementary Figure 2.

Supplementary Figure 2: Information imbalance plots obtained from independent runs of FlexConsensus trained with the helix-based synthetic dataset presented in the manuscript. The plots demonstrate that FlexConsensus executions yield similar consensus landscapes after training, indicating the reproducibility of the results.

As shown in the Figure, the information imbalance plots show that all the executions yield a value close to zero and the main diagonal. This supports that FlexConsensus is learning to produce similar consensus spaces in all cases, verifying that the landscapes obtained are reproducible.

We include below the new section added in the Supplementary Information for completeness:

FlexConsensus consistency analysis

Since FlexConsensus follows a neural network approximation, every time it is executed, it will produce a different estimate of the consensus space due to the stochastic nature of the neural network weights' optimization process. Although this will be true for all executions, it would be desirable for the minima found by the network to yield a consensus space with similar characteristics in all executions, ensuring that the network results are reproducible.

To better assess the reproducibility of the network, we conducted a test to evaluate how similar the consensus spaces predicted after training several FlexConsensus networks on the same dataset are. Following the previous idea, we trained five different consensus networks using the helix-based synthetic dataset presented in the manuscript. The overall consensus spaces predicted by the networks (i.e. the space obtained by joining the consensus space associated with each input) were then compared using the information imbalance metric:

$$\Delta(d_a \rightarrow d_B; k) = \Delta_{AB} = \frac{2}{N^2 k} \sum_{i,j}^{s.t. r_{ij}^A < k} r_{ij}^B$$

The results obtained are summarized in Supplementary Figure 2. As can be seen from the information imbalance plots, all the executions of FlexConsensus yield an information imbalance value close to zero and to the main diagonal, which verifies that the executions have yielded consensus spaces containing similar information in all cases.

6) L138: the authors make use of the Earth Mover's Distance (EMD) as a measure of accuracy of the FlexConsensus mappings. It is mentioned that this is appropriate because the input spaces can have different dimensions. However, it is not very clear from the paper why EMD is appropriate in this setting (as compared to other metrics) or exactly how the EMD is computed when the dimensionality of the inputs differ.

Since EMD measures the distance between two distributions, we can compute the similarity between two embeddings based on the EMD measure of their corresponding distance distributions. This similarity measure can be computed independently of the intrinsic dimension of the original spaces, compared to other metrics, such as the L2 norm between corresponding points, which requires the points being compared to have the same number of dimensions.

Therefore, we decided to use EMD to compute the similarity of all the embeddings involved in a given experiment. Thanks to this metric, we can compare the similarity between the input spaces and use this value to assess whether the EMD metrics computed in the consensus space are large or small.

We have further discussed all these ideas in the main text to make everything more transparent and simpler to follow. The new discussion is added below for completeness:

The decision behind choosing EMD over other distance measurements, such as the L2 norm between the spaces, is due to the possibility of computing EMD independently on the initial dimension of the spaces being compared. To achieve this, the all-to-all distances of a given space against itself are computed and used to get the distance distribution of the point composing the space. The obtained distance distributions are then used as the input for the EMD distance to measure the similarity between the input spaces. By measuring the similarities at the level of distributions, it is possible to compare the initial spaces even if they have a different dimension.

7) L172: the authors mention that the histograms in Figure 1 have a “small” error but this is not well defined. In fact it’s clear from the method that the values of error are arbitrary, based only on the scale of the input embeddings, and each input embedding can be different. If users are meant to be able to use the FlexConsensus method easily, there should be some way to tell if a certain value of error is “small”.

As the reviewer comments, the error histograms presented in the manuscript correspond to the histograms obtained from the network's representation errors (i.e., the error representing how well the decoders in the network can recover each input space from the consensus latent space). Although the distribution of these representation errors is related to the consensus, it is complicated to compare them as a consensus error metric due to their variability among different decoders and the differences in scale of the input spaces.

In general, we recommend analyzing errors directly in the consensus space. Since the inputs have been mapped to this common space, they will all have the same scale, making metrics derived from these spaces easier to interpret and compare.

Still, it is challenging to determine whether a given metric is “large” or “small,” as its size also depends on the relative scale of the consensus space itself. In the following, we present a rescaling of the consensus space error metrics that is easier to interpret in terms of “how large is large.” In this way, we decided to modify the errors computed in the consensus space according to the following formula:

$$f(x) = \frac{1}{1 + e^{-x}}$$

That is, applying a sigmoid function to map the errors to the interval $[0, 1]$. There is still an issue with the previous transformation, as the slope of the sigmoid function is set to one by default, which may lead to artefactual representations depending on the characteristics of the data. Therefore, we propose to transform these values such that $f(\mu + \sigma) = 0.5$. With this modification, the previous formula translates to:

$$f(x) = \frac{1}{1 + e^{-k(x-(\mu+\sigma))}}$$

As an additional scaling, we propose to update the slope of the sigmoid function such that $f(\mu) \approx 0.25$ and $f(\mu + 2\sigma) \approx 0.75$. If we focus on the first condition and solve for the value of k we obtain:

$$k = \frac{\ln(3)}{\sigma}$$

It is possible to show that the previous value of k also fulfills the second condition. Therefore, the sigmoid transformation proposed reduces to:

$$f(x) = \frac{1}{1 + e^{-\frac{\ln(3)}{\sigma}(x-(\mu+\sigma))}}$$

The previous transformation provides a better sense of the relative size of the errors by comparing them to the standard deviation while keeping them on a controlled range that is more robust to outliers.

We want to comment on the choice of the slope constraints selected for the sigmoid function. Although a natural choice would have been to choose $f(\mu) \approx 0.5$ instead of $f(\mu) \approx 0.25$, we have preferred the latter to give more resolution to the large error regime. In this way, it should be easier to set a threshold to filter high-consensus errors more accurately.

We have also modified the error histograms associated with the experimental datasets using the previous formula to illustrate its application in the manuscript. We have also included a new section in the Supplementary Information to describe the previous transformation.

We include below the new section for completeness:

Normalizing FlexConsensus errors

One of the main issues when working with errors and error histograms is determining whether a given error is large or small compared to the rest. This issue prevents us from easily determining which threshold should be set to the consensus to extract only those particles that reliably fulfill a given criterion.

Therefore, we propose a transformation based on a sigmoid function with additional conditions imposed. These three conditions are:

$$f(\mu) = 0.25$$

$$f(\mu + \sigma) \approx 0.5$$

$$f(\mu + 2\sigma) \approx 0.75$$

Where μ and σ are the computed errors' mean and standard deviation, respectively. The second condition can be easily achieved by shifting the sigmoid function such that:

$$f(x) = \frac{1}{1 + e^{-k(x-(\mu+\sigma))}}$$

Fulfilling the other two conditions requires finding an appropriate slope value for the sigmoid function k . If we focused on the first condition and solve for the value of k it is possible to get:

$$k = \frac{\ln(3)}{\sigma}$$

If we plug the previous value in the sigmoid equation, it is possible to check that it verifies the third condition. Thus, the final transformation reduces to:

$$f(x) = \frac{1}{1 + e^{-\frac{\ln(3)}{\sigma}(x-(\mu+\sigma))}}$$

This transformation allows the mapping of the original error values to the interval $[0, 1]$ while being robust to outliers and providing the sense of which errors are large or small by comparing them to the standard deviation of the original data.

We would like to comment on the slope constraints selected for the sigmoid function. While the natural choice would have been to set $f(\mu) \approx 0.5$ instead of $f(\mu) \approx 0.5$, we have preferred the latter to give more resolution to the largest errors. In this way, it should be easier to set a threshold to filter high consensus errors more accurately.

8) L248: the authors mention that the EMD in Figure 2b is “very large” but it is only 0.09, whereas in Figure 2a it is 0.02 - this does not seem like a substantial difference. Again if this method is to be used in the field, these values need some way to be interpretable that is clear and simple.

We agree with the reviewer that it is hard to understand if a given EMD value is large or small. Therefore, we have decided to modify them so that they are compared against the reference EMD value computed from the noiseless input spaces. This allows us to express the EMD values as a percentage, showing how big or small they are compared to this reference value.

We have modified Figures 1 and 2 to incorporate these changes and updated the main manuscript to reflect the new metrics.

9) L338: The authors mention that the figures only show a 2D projection of the 3D consensus space. Already, (as explained in point 1 above) it is not clear why the dimensionality of the consensus space should be 3, when the intrinsic dimensionality of the two input embeddings is not known. However, additionally problematic is that it is not made clear to the reader when it is okay to look at a 2D projection of the consensus space or when the 3D version (which the authors mention but don't have a figure for) is necessary. This is important because the purpose of FlexConsensus is to provide a comparison and validation tool for users, so these interpretation questions need to be clear.

The viewers implemented in Flexutils (Annotate space tool) allow direct inspection of conformational spaces using three dimensions and customizing the combination of axes displayed in each dimension.

In the case of FlexConsensus, it is also possible to visualize the consensus landscape using these tools. These tools allow one to easily visualize and extract conformational states based on the methods that initially computed the spaces (which must be selected by the user). Therefore, all the consensus spaces presented in the paper correspond to one view from the 3D visualization tool.

To clarify this issue in the main text, we have included a new paragraph at the end of the methods section describing how to use Flexutils's annotation tool to inspect the consensus space. Additionally, we have provided links to tutorials that demonstrate how to use FlexConsensus and the annotation tool to simplify their usage.

We provide below the new text added to the manuscript for completeness:

FlexConsensus as part of heterogeneity analysis workflows. The case of integration inside Scipion.

Flexibility analysis is becoming progressively a standard practice in the CryoEM field, promoted by the new emerging tools that can extract more accurately the structural information captured in the experimental images. Therefore, it is common for experimentalists to analyze the flexibility of their biomolecules with these tools. However, they are restricted in the depth of interpretation of the results reported by any given method because of the lack of validation tools and the difficulty to perform a joint analysis of the results of different methods. Our method, FlexConsensus, aims to be inserted at this point in the analysis, allowing the integration and comparison of the results yielded by different algorithms to validate whether the conformational spaces they report agree.

FlexConsensus algorithm has been integrated into Scipion as a protocol included in the Scipion Flexibility Hub (17) and accessible through the Flexutils Scipion plugin. The protocol provides a graphical interface, allowing one to easily customize the network's hyperparameters or select the neural network's inputs.

Inside Scipion, it is possible to easily execute popular heterogeneity analysis methods such as CryoDRGN (2), 3DFlex (8), or HetSIREN (15), among others. The inputs needed to execute these algorithms are also organized in a form similar to the FlexConsensus case. The particles needed to execute them can be imported into Scipion or processed inside by combining CryoEM software like Relion (18) or CryoSPARC (12). The conversions needed to adapt the outputs and inputs to different software are internally handled by Scipion, making the transition among packages smoother. These concepts also apply to the heterogeneity software integrated into Scipion and FlexConsensus.

The predictions obtained after training FlexConsensus are also automatically registered inside Scipion. This allows Scipion to communicate with the integrated visualization, analysis, and subset generation tools and gives the user more flexibility to manipulate their data according to their specific needs.

For example, users can access the 3D space annotator to view the consensus space in 3D, allowing them to select any combination of axes/dimensions to inspect and interact with it. The annotation tool also allows the recovery of real-time confirmations from any region of the consensus space to visualize them inside the annotator or using the direct connection with ChimeraX (19). Additionally, it is possible to extract the particles associated with a given region interactively and register them inside Scipion, allowing their subsequent processing in the workflow to analyze them further.

Minor comments

1) L12: experimenting should be experiencing

We thank the reviewer for finding this typo. We have corrected the typo in the abstract.

2) L80: undefined reference

We have fixed the missing reference error.

3) L307: The authors use the term “contour” and “isolines” which are somewhat confusing because in the figures, it seems that the aspect they are referring to is really the color values in the 2D histogram (i.e. the heatmap). Also the figures have dotted lines, which at first read

seem like the “contours” the authors are referring to. I think a better word could be chosen, for clarity.

Following the reviewer's suggestion, we have unified the words “contours” and “isolines” as they refer to the same information. In the updated version of the manuscript, only “subspaces” are being used to simplify the wording, making the discussion easier to follow and less confusing.

Reviewer #2

Remarks to the Author

A. Summary of the key results

Several deep-learning-based algorithms for analyzing structural heterogeneity from the cryo-EM images have been developed over the past several years. Depending on the complexity of heterogeneity (compositional or conformational heterogeneity) in the cryo-EM data, the conformational space estimations and user's interpretations of them may vary a lot. An algorithm for performing validation checks is needed in cryo-EM data processing to ensure the correct interpretation of the heterogeneous data. In this study, D. Herreros et al. developed an algorithm FlexConsensus to evaluate the agreement among the conformational landscapes generated by different existing methods. The authors built a multi-autoencoder architecture to place different conformational landscapes in a shared consensus space to compare their commonalities and differences. The authors further developed several error metrics to evaluate the reliability of the different estimations, allowing the extraction of subsets based on the consensus for further analysis.

B. Originality and significance: if not novel, please include reference

FlexConsensus is the first algorithm, as far as I know, to compare the conformational landscape estimations of cryo-EM images from different existing methods. It offers a valuable metric to assess the heterogeneity estimations from different algorithms and would benefit emerging and ongoing research focused on characterizing the dynamics of macromolecules using cryo-EM.

C. Data & methodology: validity of approach, quality of data, quality of presentation

The authors evaluate FlexConsensus's performance using both synthetic data (noise or noiseless) and real experimental data.

For the synthetic dataset, clarification is needed in the description of the data preparation: In generating the simulated data for Figure 2 (line 191 to 206), please clarify for each case (8D and 39 D), which duplicate (translated/ rotated?) has the noise added. In the Figure 2, please describe such modifications to the input spaces by adding figure legends or in the figure caption.

Following the reviewer's suggestion, we have clarified the data preparation of the second synthetic dataset presented in the manuscript to clarify the generation process.

We include below the new clarifications added to the main text and Figure 2 legend for completeness:

Focusing on the first input space, the construction starts with generating 1000 8D points, as done in the previous section. These 1000 points are then duplicated, followed by a translation of the duplicated points so they do not overlap with the original ones. This way, we obtained two replicas of the same 8D space at distinct locations. Lastly, gaussian noise was applied to the translated replica, leading to the final input space consisting of one structured region following the Lissajous curves (the original 1000 points) and another region that did not show any apparent structure due to the added high noise (the noisy duplicated 1000 points). The standard deviation of the noise added to the points was large enough to completely disrupt their structure ($\sigma = 10$).

The second input space follows the same construction process as the previous one, starting with 1000 39D points. After generating the noiseless and noisy translated replica with 1000 39D points each, the 2000 points were rotated 90 degrees around the z-axis, crossing their center of mass. This rotation simulates the randomness in the orientation of experimental spaces, as they could be randomly rotated even if the method responsible for generating that space is executed twice with the same parameters.

Figure 2: FlexConsensus analysis resulting from the comparison of two synthetic spaces with a varying number of dimensions. In addition, the synthetic spaces also mimic the presence of noise and other errors found in typical experimental latent spaces. Panel a) compares the consensus space corresponding to the 2x1000 points in the input spaces with no noise added to them. It should be noted that the consensus space was computed from the full set of 2x2000 points, including the noiseless and noisy samples. However, in this first panel, the analysis focuses only on the noiseless points highlighted in red and blue (the additional 2x1000 points coming from the noisy spaces are kept in gray and will be analyzed in the next panel). The resulting consensus landscape, error histograms derived from the representation error computed by comparing the decoded and input spaces, and the EMD_p evaluation metrics estimated from the analysis are also presented in the Figure. As done previously in the consensus spaces, the normalized error histograms highlight the errors computed from the noiseless points in red and blue. The errors associated with the noisy points are highlighted in gray and will be analyzed in the following Panel. These results show that FlexConsensus successfully identified these regions with similar structural properties. In contrast, Panel b) focuses the analysis on the 2x1000 translated points affected by noise in the input spaces. Similarly to the case before, the analysis focuses on the noisy samples highlighted in red and blue. However, the consensus space was computed from

the complete set of points, including those colored in gray. The analysis shows that FlexConsensus identifies these two regions as unreliably estimated, as they are heavily affected by the noise.

In preparing the input spaces from the experimental data, please clarify whether further pose refinement of the particles is performed by HetSIREN? If so, please comment whether this further pose refinement may contribute to the disagreement in input spaces from cryoDRGN and HetSIREN in Figure 5.

Although HetSIREN can be trained to refine the particles' pose further, it is an optional feature that can be toggled on or off directly on its Scipion protocol form.

Pose refinement was not considered to simplify data interpretation for the tests included in this manuscript.

We have further clarified this decision in the main text. An extract of this modification is included below:

It is worth mentioning that HetSIREN can further refine the poses of the images while estimating the conformational states, in contrast to CryoDRGN. However, it was decided to turn off this feature in HetSIREN so that the two methods behave similarly when estimating their conformational latent spaces, simplifying the data analysis.

D. Appropriate use of statistics and treatment of uncertainties

In Figure 6, please explain the large difference in the representation errors between the two subfigures.

The differences in the representation errors arise from the intrinsic characteristics of the input spaces estimated by HetSIREN and CryoDRGN. The relative “scale” of the vectors in the conformational landscapes estimated by different algorithms may not coincide, meaning that the average values of the latent space vector components might be larger or smaller depending on the method.

Since FlexConsensus decoders will try to map the consensus space towards a given input, each decoder will also learn to reproduce the relative scale of the latent space vector components. Thus, the relative errors obtained when decoding towards a given input space will be scaled accordingly.

We know this effect is not desirable from a practical point of view, as it will be easier to interpret the error histograms if they all have the same scale. Therefore, we have introduced a new scaling function to normalize the scale of the errors, as suggested by the first reviewer:

$$f(x) = \frac{1}{1 + e^{-\frac{\ln(3)}{\sigma}(x-(\mu+\sigma))}}$$

Thanks to the previous function, it is possible to normalize the errors to the interval [0,1] while constraining them based on the following criteria:

$$\begin{aligned}f(\mu) &= 0.25 \\f(\mu + \sigma) &= 0.5 \\f(\mu + 2\sigma) &= 0.75\end{aligned}$$

We have added an extended discussion of Figure 6 to explain the previous effect better. We provide below this new discussion for completeness (newly added text is marked in bold to highlight against the previous text in the manuscript):

To more quantitatively assess the reliability of the structural states estimated by HetSIREN and CryoDRGN, we evaluated the representation error between the input spaces and those decoded from the consensus space. Four spaces were decoded, obtained when forwarding the HetSIREN and CryoDRGN consensus spaces through the decoders responsible for generating the original two spaces from the consensus. These four spaces were then used to compute the representation errors represented as histograms in Supplementary Figure 5. The histogram analysis reveals a slight discrepancy between the two methods, as the error distributions between the spaces decoded from each consensus space do not fully overlap. This discrepancy suggests that the relative population of states estimated by CryoDRGN and HetSIREN are unequal, even though both methods correctly detect the main conformational states in the dataset. This is further supported when visually inspecting the consensus spaces included in Figure 7, as it can be seen that the overall distribution of the particles over the three main spike states is not comparable in the two methods.

It should be noted that the relative error scales will depend on the relative scale of the input space, making the comparison and interpretation of the error histograms difficult. In general, the scale of the conformational space for two different methods might not be similar, meaning that the latent vectors might be larger or smaller depending on how the method is designed. The differences in the scale of the latent vectors are transmitted through the decoders in FlexConsensus to find a mapping that is as close as possible to a given input. Therefore, the relative differences in scale are reflected in the relative representation error computed for a given input space. Thus, we applied

the normalization previously introduced in this work to simplify the comparison of the error histograms presented in Supplementary Figure 5 and improve their interpretability. This normalization makes it easier to understand the relative size of the errors while constraining them to the same range.

Figure 7, please label the Y axis of the permutation distribution plot.

We have labeled the Y axes of the permutation distribution plot as requested. We include below the new Figure for completeness (in the updated version of the manuscript, Figure 7 now corresponds to Figure 8):

E. Conclusions: robustness, validity, reliability

Please include the cost of the training in terms of computing resources and time in the methods section.

We have included training costs as requested in the main text. We include the newly added text for the completeness of this response letter:

As a reference, training the network takes around 16 seconds per epoch on an RTX Ada 6000 generation GPU, using default hyperparameters and two input spaces with 450k points each; VRAM consumption is approximately 2.6 GB.

F. Suggested improvements: experiments, data for possible revision

In this study, the authors demonstrate that FlexConsensus could access the general agreement between input spaces, allowing isolation of the outlier particles not captured by certain method e.g. deformation field-based methods to identify the compositional difference, or those noisy particles in the periphery of the shared consensus landscape as shown in Figures 3 and 5. However, another significant potential of this algorithm would be to explore the disagreement between the input spaces in the central cloud of the consensus space. For example, in the Figure 7, a large majority of particles are not with the significant consensus between two algorithms. A further analysis of such difference would shed light on the reliability of current available methods in identifying continuous heterogeneity. The authors may discuss this part in the manuscript.

Following the reviewer's suggestion, we have further discussed the results obtained in Figure 7 (which corresponds to Figure 8 in the updated version of the manuscript).

The new discussion focuses on first analyzing the relative populations obtained from the consensus space before and after filtering it; it also considers the existing large body of structural knowledge on the SARS-CoV-2 spike. The heterogeneity analysis of CryoDRGN and HetSIREN suggests that three well-defined conformational states exist in the datasets. Therefore, we carried out a 3D classification to determine a “ground truth” for the relative populations of each state.

Posteriorly, we clustered the unfiltered and filtered FlexConsensus landscapes into three groups using Gaussian Mixture models to approximate the relative populations estimated in each case. The populations obtained for each case were:

- 3D classification (Relion (19)): three down (20%), one up (60%), and two up (20%).
- HetSIREN (FlexConsensus): three down (22%), one up (50%), and two up (28%).
- CryoDRGN (FlexConsensus): three down (27%), one up (35%), and two up (38%).
- FlexConsensus filtered landscape: three down (26%), one up (41%), and two up (31%).

This result suggests that HetSIREN tends to follow a similar trend to the 3D classification, unlike CryoDRGN, which assigns a larger weight to the two-up state. The filtered landscape normalizes the original populations, decreasing the two-up states and yielding a trend that aligns with 3D classification.

Therefore, the previous tests indicate that HetSIREN and CryoDRGN estimations were intrinsically different even in the main cloud, as corroborated by visually inspecting the landscapes. Furthermore, by keeping only those images estimated similarly by both methods, FlexConsensus seems to retrieve a more normalized distribution of states closer to the 3D classification results.

In addition to the previous test, we also analyzed the 5% of images estimated to have the largest consensus error to assess whether these images correspond to unwanted images or are wrongly estimated by CryoDRGN and HetSIREN. The results are summarized in a new Supplementary Figure:

Supplementary Figure 6: Reconstruction obtained from SARS-CoV-2 dataset considering only those images estimated to be within the 5% largest consensus error (around 21k particles from the original 440k). The landscapes show the distribution of these images in the consensus space: For CryoDRGN, most of these images are located in the two-up state region, while in HetSIREN, they are distributed along the regions corresponding to the one-up and three-down states. In contrast, the reconstruction calculated directly from these particles corresponds to a clear one-up state, indicating that the two heterogeneity methods wrongly assigned these images to regions of the conformational landscape.

As the Figure shows, CryoDRGN tends to place these images around the two-up state region of the landscape, while HetSIREN distributes them along the one-up and three-down regions. Interestingly, the reconstruction obtained from these images shows a clear one-up state, confirming that these images had meaningful structural information but were wrongly positioned in conformation space by the heterogeneity methods.

We include below the new discussion added to the text for completeness:

When comparing the results obtained from the two analyses presented in Supplementary Figure 5 and Figure 8, it is possible to see that both approaches detect that the distribution of states identified by HetSIREN and CryoDRGN was incompatible in an important fraction of the images. These results suggest the importance of validating the landscapes estimated by a given algorithm to detect these inconsistencies.

However, it is interesting to understand whether the filtered FlexConsensus space yields more meaningful landscape populations. To that end, we conducted a 3D classification of the original images into several classes, which were posteriorly merged according to the RBD conformational state they represent. Since both, HetSIREN and CryoDRGN suggest a clear tendency towards the presence of three well-differentiated conformational states, the populations of the 3D classification will be considered as the "ground truth" populations as they are obtained from a standard approach in the field.

In the case of the landscapes, Gaussian Mixture-based clustering was used to compute the populations. To that end, three Gaussian components were estimated to cluster the spaces to get the relative populations to compare against the 3D classification ones. It should be noted that the populations were directly measured in the consensus subspaces of HetSIREN and CryoDRGN, which were obtained using FlexConsensus.

The relative populations obtained were:

- 3D classification (Relion (19)): three down (20%), one up (60%), and two up (20%).
- HetSIREN (FlexConsensus): three down (22%), one up (50%), and two up (28%).
- CryoDRGN (FlexConsensus): three down (27%), one up (35%), and two up (38%).
- FlexConsensus filtered landscape: three down (26%), one up (41%), and two up (31%).

By comparing the different populations against the 3D classification, it is possible to see how HetSIREN tends to be more conservative, estimating a set of populations that follow a similar trend to the 3D classification ones (i.e., a larger number of one-up states compared to the other two). In contrast, the trend of CryoDRGN is to give a larger population to the two-up states. Flexconsensus filtered landscape tends to normalize the populations to a value between the two original methods, leading to a new set resembling the behavior seen in the classification results.

A question remains on analyzing the characteristics of those images estimated to have a large consensus error. It is important to understand whether they correspond to "bad" particles or, instead, their position in conformational space arises from estimation errors from the heterogeneity methods. Therefore, an additional experiment was conducted to compare the reconstructions obtained from those images estimated to have a consensus error larger than a given threshold. The results obtained in this analysis are summarized in Supplementary Figure 6. As shown in the Figure, and as expected, these particles are distributed differently in the HetSIREN and CryoDRGN consensus spaces. In CryoDRGN, the images are placed close to the two-up region, while HetSIREN tends to distribute them along the one-up and three-down states. In contrast, the volume reconstructed from these images yields a clear one-up state, indicating that the quality of these images was not intrinsically bad but that, for yet unknown reasons, both HetSIREN and CryoDRGN did not properly position these images in conformational space close to other images in similar states of RBD opening. Consequently, although the conceptual distinction between images with large consensus errors being either intrinsically bad or badly estimated by a method is very important, it seems we do not have a clear way to distinguish between these two options.

G. References: appropriate credit to previous work?

Yes.

H. Clarity and context: lucidity of abstract/summary, appropriateness of abstract, introduction and conclusions

The manuscript is well structured and written. A few typos and errors need to be corrected.

Figure 3b caption: For Zernike3D, the highlighted regions correspond to the region where HetSIREN has detected the biggest conformational variability change. Do the authors mean compositional variability?

As the reviewer suggests, the caption of Figure 3b (now Figure 6c in the updated manuscript) contained a typo. We have corrected this mistake in the new version of the manuscript.

Typos in Line 314 and 449.

We thank the reviewer for noticing these two typos, which we have corrected in the new version of the manuscript.

Reviewer #3

Remarks to the Author

Summary of the paper

This manuscript introduces FlexConsensus, a novel multi-autoencoder architecture that merges conformational landscapes from different heterogeneity analysis algorithms in cryo-electron microscopy (cryo-EM). The authors demonstrated that FlexConsensus can:

1. Create a unified consensus space from disparate conformational landscapes with different dimensionalities.
2. Allow for the identification and filtering of particles based on their stability across different algorithms.
3. Enable conversion between different conformational landscape representations.

The methodology is validated using both synthetic datasets with controlled heterogeneity and real experimental datasets (EMPIAR 10028 and SARS-CoV-2 D614G spike). Overall, FlexConsensus represents a significant contribution to the field of cryo-EM heterogeneity analysis by providing a framework to compare and integrate results from different algorithms.

Suggested improvements

Consideration of related approaches

The manuscript would benefit from a brief discussion of related methods that address different aspects of heterogeneity analysis:

1. "Integrating molecular models into CryoEM heterogeneity analysis using deep neural networks" [1] incorporates prior structural knowledge from molecular modeling. While FlexConsensus focuses on comparing methods, incorporating structural priors could further enhance its biological interpretability.
2. "OPUS-DSD: deep structural disentanglement for cryo-EM" [2] takes a different approach by disentangling complex structural variations into interpretable components. FlexConsensus might benefit from exploring whether consensus spaces could similarly be disentangled into biologically meaningful dimensions.

3."Cryo-EM heterogeneity analysis using regularized covariance estimation" [3] employs statistical frameworks for heterogeneity analysis. FlexConsensus could potentially incorporate similar statistical measures to provide confidence metrics for its consensus representations.

A high-level discussion of how FlexConsensus relates to these alternative approaches would provide valuable context for readers and highlight the unique contributions of this method within the broader landscape of heterogeneity analysis tools.

Following the reviewer's suggestion, we have included a brief discussion in the Introduction to compare FlexConsensus against the proposed methods, focusing on its unique characteristics.

We include below the new paragraph added to the Introduction for completeness:

FlexConsensus proposes a framework that is not aimed at estimating states from CryoEM data but focuses on mapping already estimated landscapes into a common consensus space while preserving the organization of the original landscapes and enhancing their interpretability. Therefore, the application of FlexConsensus does not involve computing or modifying a given conformational space with a priori information, as suggested by other methods, such as the incorporation of structural priors (6), the disentanglement of the space into meaningful components (4), or methods employing statistical approaches to analyze the landscape (7), among others. Instead, analyzing the consensus spaces facilitates the identification of similarities and differences among various methods, assisting in the validation of estimations and providing tools to streamline the heterogeneity workflow based on the derived reliability scores.

Algorithm details and estimation

1.The implementation details of the multi-autoencoder architecture could be more thoroughly described. Specific network architecture choices (number of layers, neurons per layer, activation functions) and hyperparameters (learning rate, batch size, regularization parameters, optimizer choice) should be included

Following the reviewer's suggestion, we have thoroughly detailed the multi-encoder architecture and indicated the network's hyperparameters.

We include below the newly added text for completeness.

Each input conformational landscape is forwarded through its encoder, which is composed of 3 fully connected layers with 1024 neurons each and ReLU activation. The output layer of every encoder is then fed to a standard linearly activated latent space layer of a variable number of neurons that the

user can choose. This bottleneck will become the common conformational space after the network is trained.

Lastly, each encoded data is forwarded through the corresponding decoder, trying to restore the original conformational space based on the information of the common latent space. Similarly to the encoders, each decoder comprises a set of 3 fully connected layers with 1024 neurons each and ReLU activation, followed by a final layer with the same dimension as the corresponding initial conformational space and linear activation.

The network is trained using an Adam optimizer. By default, the learning rate is set to $1e^{-5}$, and the batch size is 1024. However, users can modify these hyperparameters to adjust them to their specific needs.

As a reference, training the network takes around 16 seconds per epoch on an RTX Ada 6000 generation GPU, using default hyperparameters and two input spaces with 450k points each; VRAM consumption is approximately 2.6 GB.

2.The manuscript would be significantly strengthened by implementing simpler baseline methods for comparison. A systematic benchmarking against classical dimensionality reduction techniques such as t-SNE and simple distance-based clustering of conformations.

Although FlexConsensus could be used as a dimensionality reduction method, it is not its primary purpose, as it cannot be executed with a single space. Nevertheless, prompted by the referee's suggestion, we have decided to include the comparison proposed by the reviewer in the Supplementary Information and a new Supplementary Figure showing how the displayed spaces in the experiments compare to the t-SNE and distance-based clustering representation of the original spaces. We hope this comparison can show how the different methods compare on dimensionality reduction tasks.

We include below the new discussion for completeness:

Comparison of FlexConsensus against popular dimensionality reduction methods

Although the primary purpose of FlexConsensus is to find the mapping that allows the merging of different conformational spaces into a common consensus space, it also reduces the dimensionality of the input data based on its different cost functions and regularizations. Therefore, it is interesting to assess how the dimensionality reduction capabilities of FlexConsensus compare to other standardized methods such as t-SNE or distance-based clustering.

To simplify the comparison, we decided to compare the FlexConsensus spaces obtained from HetSIREN for the EMPIAR-10028 and SARS-CoV-2 datasets presented in the manuscript against the dimensionality-reduced version yielded by t-SNE and distance-based clustering from the original HetSIREN latent space. The previous comparison is summarized in Supplementary Figure 1. As can be seen from the comparison, FlexConsensus and t-SNE tend to reproduce the different local features of the landscape better than distance-based clustering.

For the EMPIAR-10028 dataset, both FlexConsensus and t-SNE properly capture the three main regions detected by HetSIREN, corresponding to the three main conformational states it detects. In contrast, distance-based clustering only detects two clouds, as highlighted in the image with the dotted ellipses.

In the SARS-CoV-2 landscape, the differences between FlexConsensus, t-SNE, and distance-based clustering are more apparent. The first two are the only ones able to detect the three main states found by HetSIREN properly.

Supplementary Figure 1: Comparison of the dimensionality reduction capabilities of FlexConsensus against t-SNE and distance-based clustering. For the two datasets presented, the original HetSIREN executions analyzed in the manuscript were used as the input for each algorithm.

Specific experimental additions

1.Resolution assessment: Analyze whether particles identified as having high consensus error affect the resolution of reconstructed volumes. Apply different consensus error thresholds

(removing 5%, 10%, 15% of high-error particles), perform 3D reconstruction, calculate FSC curves, and report resolution changes at different filtering levels.

As suggested by the reviewer, we have performed the proposed analysis to study how the resolution of a reconstruction changes when particles are filtered according to the consensus scores. The following image summarizes the results obtained for the EMPIAR 10028 and SARS-CoV-2 datasets presented in the manuscript:

As can be seen from the FSC curves, the volumes obtained after filtering at different levels are very similar. This interesting finding suggests that the particles detected to have a larger consensus error do not correspond to "wrong" images but to wrong estimations of the variability captured in an image by the initial methods.

To prove this, we analyzed the "negative" subsets of images (i.e., the subset of images with a larger consensus error). The results from this analysis are summarized in Supplementary Video 3 (Empiar-10028 dataset) and Supplementary Figure 6 (SARS-Cov-2 dataset). These new results confirm that the images detected to have a larger consensus error have meaningful structural information but are wrongly estimated by the original methods, thus translating into a larger consensus error.

This result highlights the importance of having tools to interact with the data. These tools allow for the easy analysis of different subsets of images using consensus metrics to understand better how the estimations of different methods compare.

2.1 recommend comparing FlexConsensus with standardized benchmark datasets from CryoBench [4] to evaluate performance on well-characterized heterogeneity challenges better:

IgG-RL dataset: This dataset models random configurations of an antibody linker peptide, creating complex, non-linear conformational variability. Testing FlexConsensus on this dataset would validate its ability to handle disordered regions and multiple conformational degrees of freedom, particularly demonstrating whether the consensus space can effectively capture the relationship between center-of-mass distance and dihedral angle parameters that characterize the ground truth. The authors should specifically analyze whether the consensus space dimensions correlate with the physical parameters of the linker motion, and whether particles with high consensus error correspond to conformational extremes or artifacts.

Spike-MD dataset: Derived from molecular dynamics simulations of SARS-CoV-2 spike protein (46,789 unique conformations), this dataset would challenge FlexConsensus with subtle, continuous conformational changes across multiple scales. This test would demonstrate whether the consensus metrics can identify meaningful biological transitions such as the receptor binding domain opening, and whether the consensus space dimensions correlate with the established collective variables (CV1/CV2) that parameterize the conformational landscape. The authors should map the consensus space onto known energy landscapes from the MD simulation to assess whether the method captures energetically relevant states and transitions.

For more rigorous validation, data from CryoBench could serve as a reference for performance assessment. The authors could select relevant datasets for further validation and quantify performance using established metrics like neighborhood similarity and information imbalance.

Following the reviewer's suggestion, we have included in the manuscript a new section with a thorough analysis of the IgG-RL and MDSpoke datasets included in CryoBench. Before discussing the results, we wanted to comment on the datasets available online, which were detected to have some errors in the star files. These errors were mostly related to metadata that was not appropriately set, like the sampling rate. Moreover, the star files were only compatible with Relion 3, making it impossible to import them with newer versions of Relion, like version 4 or 5. Therefore, our first step in the analysis was to carefully evaluate the datasets to correct these inconsistencies and generate new star files compatible with newer versions of Relion.

Once the two proposed datasets were corrected and tested, we analyzed them to ensure they worked properly. In both experiments, we used HetSIREN (with no pose refinement) and CryoDRGN to get two estimations of the conformational landscapes to be input to FlexConsensus.

Starting with the IgG-RL dataset, we first performed an analysis similar to the ones carried out in other datasets, as shown in the Figure below and in Supplementary Video 1:

Figure 4: FlexConsensus analysis resulting from comparing the estimations of HetSIREN and CryoDRGN for the IgG-RL dataset included in CryoBench (10). The image shows both the original consensus space derived from the two methods and the consensus space obtained by keeping only those particles in the consensus space that are significantly similar. The first two landscapes displayed in the image correspond to the unfiltered consensus spaces obtained with FlexConsensus. Following these spaces, we include the plot with the different p-values obtained from the permutation test when filtering an increasingly larger number of samples, highlighting the threshold proposed by the test. Below the previous plot, the permutation distribution of distances for the selected threshold is included, showing that most of the measurements computed from the random labeling fall below it. Lastly, the Figure shows the consensus spaces obtained after applying the threshold proposed by the test, showing a more comparable distribution of states.

The analysis also includes the consensus space's filtering process to retain only particles whose conformation was estimated to be significantly similar. Comparing the original consensus spaces

shows that the distribution of states found by each method is quite different, which is a characteristic of this dataset already noted in the original publication of CryoBench. To assess these differences more quantitatively, we performed the statistical analysis proposed in the manuscript to automatically detect and filter those significantly different images according to the consensus estimations. This test shows that a large fraction of the input images are discarded according to this criterion, confirming that the conformational spaces estimated by HetSIREN and CryoDRGN were significantly different. This situation makes the role of a method like FlexConsensus especially clear and valuable, as it can simultaneously consider the results of several methods.

To better characterize the correlation of the filtered spaces with the antibody motions present in the dataset, we carried out an additional experiment as proposed by the reviewer. This experiment begins by clustering the filtered consensus space into 10 distinct groups, enabling the extraction of cluster representatives. The representatives were then converted back to HetSIREN latent vectors using the decoding capabilities of FlexConsensus to decode the different conformational states associated with them. These states were then segmented using Segger to isolate the heavy and light chains of the protein, allowing for the measurement of three different centers: the center of mass of the heavy chain, the center of mass of the light chain, and the anchor point between the two chains. Thanks to these centers, it is possible to characterize the motion of the protein by measuring the distance between the two aforementioned centers of mass and the angle described by the previous three centers.

The next step is to determine whether the previous two variables (distance between centers of mass and angle) are correlated with the location in the filtered consensus space. To that end, we performed a Mantel test under the hypothesis that the distance matrices defined by the consensus latent vectors at the variables measured in the volumes are correlated, yielding a p-value of 0.002. This result confirms a strong correlation between the position in the consensus space and the conformational changes present in the dataset. This experiment is summarized in Supplementary Figure 7:

Supplementary Figure 7: Evaluation of the correlation between the FlexConsensus latent space vectors obtained for the IgG-RL dataset from CryoBench (1) and two internal variables (d and θ) designed to characterize simulated disordered motions.

The second dataset analyzed was the MDspike dataset. The results obtained from the FlexConsensus analysis of HetSIREN and CryoDRGN are summarized in the following Figure and Supplementary Video 2:

Figure 5: FlexConsensus analysis resulting from comparing the estimations of HetSIREN and CryoDRGN for the MDSPike dataset included in CryoBench (10). The image shows both the original consensus space derived from the two methods and the consensus space obtained by keeping only those particles in the consensus space that are significantly similar. The first two landscapes displayed in the image correspond to the unfiltered consensus spaces obtained with FlexConsensus. Following these spaces, we include the plot with the different p-values obtained from the permutation test when filtering an increasingly larger number of samples, highlighting the threshold proposed by the test. Below the previous plot, the permutation distribution of distances for the selected threshold is included, showing that most of the measurements computed from the random labeling fall below it. Lastly, the Figure shows the consensus spaces obtained after applying the threshold proposed by the test, showing a more comparable distribution of states.

In this case, the landscape analysis reveals that the estimations of HetSIREN and CryoDRGN are more similar, resulting in a lower percentage of particles being discarded by the filtering process. Additionally, Supplementary Video 2 demonstrates that the consensus landscape accurately captures the one-to-three-down transition simulated in this dataset.

To better characterize the consensus' ability to capture relevant structural information, we conducted an additional experiment to determine whether the ground-truth energy landscape and its associated collective variables are correlated with the consensus, as suggested by the reviewer. Since we know the correspondences between the MDSPike ground truth latents and the estimated ones, we can use FlexConsensus to map the three spaces in the consensus space together. For this case, we set the consensus space to have two dimensions, matching the ground truth space. Once the network was trained, we extracted several latent conformations along the two principal axes of the ground truth space, allowing us to define our two collective variables. These latent conformations were then mapped with the trained network in the consensus space to check their correlation with the consensus space axes. The results from this analysis are summarized in Supplementary Figure 8:

Supplementary Figure 8: Comparison of the location of the collective variables measured from the ground truth energy landscape of the MDSPike dataset from CryoBench (1) in its original energy landscape and FlexConsensus space. Three

landscapes were input to the consensus: the ground truth energy landscape, the conformational landscape of HetSIREN, and the conformational landscape of CryoDRGN. The results highlight the correlation of the original collective variables (CV1 and CV2) with the consensus space, which are almost aligned with the consensus space's principal components (PC).

As seen from the Figure, the result confirms that a strong correlation exists between the collective variables measured in the ground truth energy landscape and the axes of the consensus space, showing the method's ability to capture relevant structural information from the simulated motions.

We include below the new section added to the manuscript for completeness:

Consensus results on CryoBench datasets

Before moving to testing FlexConsensus on experimental data, it is interesting to test its performance on controlled and challenging datasets capturing different ranges of motion and variability. Therefore, we tested FlexConsensus on two different datasets included in CryoBench (10).

The first dataset to be explored is the IgG-RL one, representing a set of random conformations for a disordered peptide linker connecting the Fab to the rest of the IgG complex. This dataset corresponds to a challenging disordered motion, which is very difficult to follow by all proposed heterogeneity analysis methods so far, as indicated in the original CryoBench manuscript. Indeed, the conformational landscapes obtained by the different methods have significant differences. It is an excellent example for integrative tools like FlexConsensus, where we can explore the similarities and differences among different methods.

The proposed experiment starts by estimating the conformational spaces using two different methods: HetSIREN and CryoDRGN. To simplify the FlexConsensus results analysis, HetSIREN was forced to estimate only a conformational latent space without refining the original angles, so its behavior is similar to CryoDRGN. The landscapes predicted by each method were then fed into FlexConsensus to estimate the consensus landscape. The results from this analysis are summarized in Figure 4. The consensus space obtained from the two original estimations shows a clear difference in the distribution of states found by the two methods. This can be further quantified by analyzing the consensus errors measured in the consensus space.

Thus, consensus metrics from the previous analysis were used to filter the landscape toward a stabilized representation with a more reliable state distribution. Based on the consensus, it is possible to define a statistical framework to determine a significant threshold based on the similarity of the distribution of states represented by the different consensus landscapes. To that end, it is possible to

work under the assumption that the distributions of states of different methods in the consensus space should be the same, allowing us to derive the threshold that fulfills the previous assumption.

Following the previous reasoning, we proposed an approach based on FlexConsensus to derive the previous threshold based on a random permutation test of the distances between identical samples in the consensus space. It allows us to obtain a p-value representing the probability of measuring the distances observed in the consensus space against random labeling. The Methods section describes this filtering process and the threshold computation from the previous p-value in more detail.

The application of the previous test to the IgG-RL consensus space is summarized in Figure 4. At the beginning of the Figure, we show the unfiltered consensus spaces followed by the plot of the p-values the previously described test yielded. Additionally, we show below the previous plot the permutation distribution of distances for the selected threshold, showing that most of the measurements computed from the random labeling fall below it. After applying the threshold highlighted in these plots, it is possible to get a representation of the consensus space, including only those images estimated to have a similar conformational state according to the two input methods.

For this experiment, the filtering process suggests that the distribution of states in the original spaces was not similar, which aligns with the findings found in the original CryoBench paper (10). This is shown when checking the threshold found with this test as presented in the "P-Value analysis" plot, which translates into a large fraction of discarded images (only around 10k images are kept after applying the threshold). Additionally, the filtered space isolates more independent antibody states than the original consensus, helping identify the different motions simulated in the dataset. In Supplementary Video 1, we inspect the filtered consensus space and the associated conformational states. To better quantify the relationship between the antibody's position and the location in the filtered consensus space where potentially different conformational states are more easily detected, we performed an additional analysis to determine if the previous two variables are correlated. To that end, the filtered consensus space was split into ten uniformly distributed clusters with KMeans. The representatives of each cluster were then converted to HetSIREN latent space vectors using the decoding capabilities of the FlexConsensus network, allowing us to recover the conformational states associated with the cluster representatives. Each volume was segmented using Segger (11) to isolate the heavy and light chains of the protein. Thanks to this segmentation, it is possible to measure the center of mass of the fixed heavy chain and the center of mass of the light chain at the different conformations. To characterize the motion of the antibody, we measured the distance from the center of mass of the heavy chain to the center of mass of the light chain, as well as the angle defined by the previous two centers of mass and the anchor point between the heavy and the light chains.

The next step was to characterize the presence of a correlation between the two measured variables (distance between centers of mass and angle) and the location in the filtered FlexConsensus space. To that end, we carried out a Mantel test working under the hypothesis that the distance matrices defined by the FlexConsensus latent vectors and the two variables measured in the volumes are correlated. A summary of this experiment is provided in Supplementary Figure 7. As seen from the Figure, the Mantel test returns a p-value of 0.002, confirming the hypothesis that there is a correlation between the motions of the antibody at the level of maps and the location in the consensus space.

The next second dataset explored was the MDSPike dataset included in CryoBench. This dataset was simulated from a long-timescale molecular dynamics simulation, yielding a free-energy landscape that was sampled to generate approximately 50000 structures. These structures were then converted to electron density maps and projected to create the images in the dataset. The ground-truth locations of these structures in the energy landscape are also provided.

Similar to the case before, HetSIREN (with no pose refinement) and CryoDRGN were executed to estimate the conformational space from this dataset, which was then analyzed with FlexConsensus. The resulting consensus spaces are shown in Figure 5. As can be seen from the Figure, the distribution of states estimated by the two methods was more similar in this case, which also translates into a filtered landscape losing a lower fraction of particles. In Supplementary Video 2, we provide an inspection of the filtered consensus space and the associated conformational states, which shows the transition from the one-up RBD state to the three-down state simulated in this dataset.

Apart from comparing estimations from different methods, the MDSPike dataset opens an interesting possibility for evaluating how methods compare to real energy landscapes. Therefore, we perform additional analysis to map experimental and simulated landscapes in a common space, focusing on the possibility of correlating collective variables characteristic of a given motion with the consensus space. To that end, we trained a new FlexConsensus network with the ground truth latent coordinates obtained from the simulation and the experimental landscapes obtained with HetSIREN and CryoDRGN. Since the ground truth space was defined to have two dimensions, FlexConsensus latent space was set to have this number of dimensions.

Once the network was trained, we took different latent vectors in the ground truth space along the two principal axes of the energy landscape, allowing us to define our collective variables. The extracted data was then input to FlexConsensus to map them in the consensus space, allowing us to evaluate their correlation with the consensus axes. The results of this experiment are summarized in Supplementary Figure 8. As seen from the Figure, the collective variables extracted from the principal

axes of the ground truth space have a strong correspondence with the main axis of the consensus space, implying that the consensus space has been able to capture the relevant structural information from the simulated motions.

Figure improvements

1. The uploaded figures are not vector graphics, causing pixelation when zoomed.

We have submitted the Figures as PPT files to increase the resolution of vector graphics. We have also increased the resolution of bitmaps in the PPT files, allowing them to be zoomed to a greater degree. Additionally, we have increased the resolution of the images included in the manuscript.

2. Add visualization comparisons of the consensus space against simpler approaches such as t-SNE and simple distance-based clustering of conformations.

As mentioned in our response letter, we have included a new Supplementary Figure with the suggested comparison.

User accessibility

More discussion on how structural biologists can practically incorporate this tool into their workflow would enhance the manuscript's impact.

Following the reviewer's suggestion, we have further discussed in a new Method subsection how FlexConsensus can be accessed through Scipion to define a heterogeneity analysis workflow, including the new consensus step. In addition, in the Code Availability Section, we have included a link to the tutorials we have prepared to show how FlexConsensus and other heterogeneity tools like CryoDRGN or 3DFlex can be executed and incorporated into a complete workflow.

We include below the new subsection added to the main text for completeness:

FlexConsensus as part of heterogeneity analysis workflows. The case of integration inside Scipion

Flexibility analysis is becoming progressively a standard practice in the CryoEM field, promoted by the new emerging tools that can extract more accurately the structural information captured in the experimental images. Therefore, it is common for experimentalists to analyze the flexibility of their biomolecules with these tools. However, they are restricted in the depth of interpretation of the results reported by any given method because of the lack of validation tools and the difficulty to perform a joint analysis of the results of different methods. Our method, FlexConsensus, aims to be

inserted at this point in the analysis, allowing the integration and comparison of the results yielded by different algorithms to validate whether the conformational spaces they report agree.

FlexConsensus algorithm has been integrated into Scipion as a protocol included in the Scipion Flexibility Hub (17) and accessible through the Flexutils Scipion plugin. The protocol provides a graphical interface, allowing one to easily customize the network's hyperparameters or select the neural network's inputs.

Inside Scipion, it is possible to easily execute popular heterogeneity analysis methods such as CryoDRGN (2), 3DFlex (8), or HetSIREN (14), among others. The inputs needed to execute these algorithms are also organized in a form similar to the FlexConsensus case. The particles needed to execute them can be imported into Scipion or processed inside by combining CryoEM software like Relion (18) or CryoSPARC (12). The conversions needed to adapt the outputs and inputs to different software are internally handled by Scipion, making the transition among packages smoother. These concepts also apply to the heterogeneity software integrated into Scipion and FlexConsensus.

The predictions obtained after training FlexConsensus are also automatically registered inside Scipion, allowing it to communicate with the integrated visualization, analysis, and subset generation tools and giving the user a larger flexibility to manipulate their data according to their specific needs.

For example, users can access the 3D space annotator to view the consensus space in 3D, allowing them to select any combination of axes/dimensions to inspect and interact with it. The annotation tool also allows the recovery of real-time confirmations from any region of the consensus space to visualize them inside the annotator or using the direct connection with ChimeraX (19). Additionally, it is possible to extract the particles associated with a given region interactively and register them inside Scipion, allowing the processing of them posteriorly in the workflow to analyze them further.

The new Code Availability section reads as follows:

The FlexConsensus algorithm is available through Scipion 3.0 (14) under the plugins and *scipion-em-flexutils* (21) (<https://github.com/scipion-em/scipion-em-flexutils>) and the package *Flexutils-Toolkit* (22) (<https://github.com/I2PC/Flexutils-Toolkit>). The protocol corresponding to the algorithm described in this manuscript is *flexutils - train - FlexConsensus* and *flexutils - interactive consensus - FlexConsensus*.

Tutorials on how to set up and use FlexConsensus are provided on the following webpage: https://scipion-em.github.io/docs/release-3.0.0/docs/user/tutorials/flexibilityHub/main_page.html\#tutorials

In addition, we provide an interactive Scipion workflow visualization showing an example of how to execute FlexConsensus here:

<https://scipion.i2pc.es//cryoemworkflowviewer/workflow/aac223841371a67510e9eab16ac8870246b30d68>

[1] Chen M, Toader B, Lederman R. Integrating molecular models into CryoEM heterogeneity analysis using scalable high-resolution deep Gaussian mixture models[J]. *Journal of molecular biology*, 2023, 435(9): 168014.

[2] Luo Z, Ni F, Wang Q, et al. OPUS-DSD: deep structural disentanglement for cryo-EM single-particle analysis[J]. *Nature Methods*, 2023, 20(11): 1729-1738.

[3] Gilles M A, Singer A. Cryo-EM heterogeneity analysis using regularized covariance estimation and kernel regression[J]. *Proceedings of the National Academy of Sciences*, 2025, 122(9): e2419140122.

[4] Jeon M, Raghu R, Astore M, et al. CryoBench: Diverse and challenging datasets for the heterogeneity problem in cryo-EM[J]. *Advances in Neural Information Processing Systems*, 2024, 37: 89468-89512.

Remarks on code availability

Code is available through Scipion 3.0 via `scipion-em-flexutils` plugin with specific FlexConsensus protocols.

RESPONSE LETTER

In the following document, we include our detailed answers to the reviewer's comments, including a description of the modifications in the manuscript to simplify their tracking in the new version. The letter is formatted so the reviewer's comments are highlighted in blue, followed by the answers in dark green. For the completeness of the response letter, we also include the original text added in the reviewed version of the manuscript in purple at the end of every answer.

Reviewer #1

Response to the author's rebuttal

1)

a) While I feel that the discussion of intrinsic dimensionality of the latent spaces is still very important for a method like FlexConsensus, the change of the default to use the min of input dimensions is a good step.

There may be a typo, but the rebuttal document says that the default is “automatically set the number of dimensions to the minimum of all the input embeddings”, but new manuscript text still says that the default is three. This should be corrected.

We thank the reviewer for finding this typo in the manuscript. We have corrected it in the new version of the manuscript.

We include, following the comment below, the new text added to the manuscript to comply with the two changes suggested by the reviewer.

Given that the purpose of FlexConsensus is not just to approximately match the input latent embeddings to each other, but to measure and compare them, I think it's important that another sentence be added to the manuscript to the effect that “Setting a consensus space dimension smaller than the intrinsic dimensionality of the input latent dimensions means that some information from the inputs will potentially be ignored when mapping to the consensus space.”

We agree with the reviewer that further clarification on the effect of setting a small dimension in the consensus space is helpful. We have added the proposed sentence in the new version of the manuscript.

We include below the new text added to the manuscript for completeness purposes:

We note that FlexConsensus learns by default a consensus space with the number of dimensions set to the minimum dimension among the input spaces. However, the latent space dimensionality is exposed as a customizable parameter in the corresponding Scipion (14) protocol form. It is important to note that setting a consensus space dimension smaller than the intrinsic dimensionality of the input latent dimensions means that some information from the inputs will potentially be ignored when mapping to the consensus space.

All the consensus landscapes presented in the following sections were defined to have three dimensions, thereby avoiding the need for any further dimensionality reduction step.

2)

In the rebuttal text, the authors explain that the methods that produced the input spaces may have failed to capture the variation of the particle; but that is not what this point was about. The point is that the synthetic test cases in the original draft were really not complex or difficult enough to provide evidence that the method works.

The new synthetic example is better than the original examples - the intrinsic dimensionality of one of the inputs is at least greater than 1, and the two inputs do not have any identical dimensions. The mapping to consensus space is clearly non-linear. So it is a much better test in my opinion and I think helps improve the paper.

This new synthetic test in combination with the examples on real data I think are now sufficient for readers to be confident in the method.

3)

a) Thank you, I can agree with the author's argument.

b) c) The new text and experiment is useful. Showing additional analysis using the reproducibility as defined as consensus error helps to confirm that in practice it measures something that is meaningful for the dataset.

The additional comparison with 3D classification is also helpful to make concrete the meaning of error in the consensus space.

The added text also helps the reader to understand the difference between “bad” particles and estimation errors in the input heterogeneity methods.

4) Thank you for making the suggested change.

5) Thank you, the addition to the supplementary materials helps clarify.

6) Thank you, the additional text helps clarify.

7) Thank you, the new method for normalizing makes comparisons of size more clear.

8) Thank you for the change.

9) Thank you, the new section about the workflow is an important and useful addition.

Minor comments all addressed.

Based on the changes, I would be happy to see the paper proceed.

Reviewer #2

Remarks to the Author

The authors have addressed the previous concerns raised.

A few more questions need to be clarified in the manuscript.

1. Supplementary Figure 6 shows that the particles with higher consensus errors are not necessarily "bad" particles; rather, they may represent images that are incorrectly assigned in the conformational landscape by individual heterogeneity methods. This raises the question: if these particles are more accurately estimated in a high-dimensional conformational space, could they be mistakenly assigned when projected into a lower-dimensional consensus space? In other words, how much does the choice of consensus space dimensionality influence the distribution of consensus errors? The author may need to clarify this in the manuscript.

As the reviewer suggests, if the dimensionality of the consensus space is smaller than that of the original spaces, there will be an intrinsic loss in accuracy due to placing high-dimensional points on a lower-dimensional one. Although the network will attempt to minimize this error as much as possible, some particles may be more affected by this accuracy loss than others.

Therefore, following the first reviewer's suggestion, we have set the default consensus space dimensionality to the minimum dimensions among the input spaces, thereby minimizing the previous effect as much as possible. Additionally, we have explicitly commented on the impact of setting a small number of dimensions in the consensus space in the manuscript to clarify it for readers.

The new text added to the manuscript is included below for completeness purposes:

We note that FlexConsensus learns by default a consensus space with the number of dimensions set to the minimum dimension among the input spaces. However, the latent space dimensionality is exposed as a customizable parameter in the corresponding Scipion (14) protocol form. It is important to note that setting a consensus space dimension smaller than the intrinsic dimensionality of the input latent dimensions means that some information from the inputs will potentially be ignored when mapping to the consensus space.

All the consensus landscapes presented in the following sections were defined to have three dimensions, thereby avoiding the need for any further dimensionality reduction step.

2. In the revised manuscript, lines 57–65, the authors discuss the current limitations of validation of heterogeneity methods. Later, in lines 938–942, they introduce the 3D space annotator, which is used to generate real-time conformations from any region within the consensus space. However, clarification is needed regarding whether the 3D space annotator outputs a single volume per particle with high confidence. Specifically, is the interpretability of the output still constrained by the number of particles surrounding the center of a given class?

The space annotator introduced in the manuscript relies on a given method to produce the latent space being visualized and decode a given volume based on any latent vector belonging to that space. Therefore, the annotator is not performing a reconstruction with a subset of particles surrounding a given latent vector, but instead retrieving the volume associated with that exact vector using a specific method.

In the case of FlexConsensus, the annotator will receive both the consensus space (which is the one to be displayed) and the method to be used for decoding a given volume. Then, we can use the decoder from FlexConsensus or a simple association of the consensus latent vectors to recover the original latent vectors, which are then fed into the decoder of the selected method to recover the decoded volume from a given latent.

We have further clarified this in the main text. We include below the new discussion for completeness:

For example, users can access the 3D space annotator to view the consensus space in 3D, allowing them to select any combination of axes/dimensions to inspect and interact with it. The annotation tool also enables the recovery of real-time conformations from any region of the consensus space to visualize them inside the annotator or using the direct connection with ChimeraX (20). To extract the previous conformations, the annotator relies on the specific method that generated the latent space being analyzed to recover the same state seen by that method for any location in the conformational space. This avoids the need to reconstruct with a neighborhood of particles surrounding a given point in the conformational space.

Reviewer #3

Remarks to the Author

A. Summary of the key results

The authors have successfully addressed all my concerns from the first review. The revised manuscript now presents FlexConsensus as a comprehensive tool for merging conformational landscapes from different cryo-EM heterogeneity analysis methods, with robust validation on both synthetic and experimental datasets.

B. Originality and significance

The significance of this work remains high. The additional experiments on CryoBench datasets further strengthen the method's validity and demonstrate its broad applicability to challenging heterogeneity problems in cryo-EM.

C. Data & methodology: validity of approach, quality of data, quality of presentation

- The authors have significantly improved the methodology section with following:
 - Detailed network architecture description (3 fully connected layers, 1024 neurons each, ReLU activation)
 - Clear hyperparameter specifications (learning rate 1e-5, batch size 1024, Adam optimizer)
 - Computational requirements (16 seconds/epoch on RTX Ada 6000 GPU, ~2.6GB VRAM)
 - Comprehensive comparison with t-SNE and distance-based clustering (Supplementary Figure 1)

D. Appropriate use of statistics and treatment of uncertainties

The statistical analyses are now robust, particularly the Mantel test for IgG-RL dataset ($p=0.002$) and the correlation analysis with collective variables for the MDSpoke dataset.

E. Conclusions: robustness, validity, reliability

The conclusions are well-supported by the expanded experimental evidence. The finding that high consensus error particles are not "bad" particles but rather incorrectly estimated by the methods is particularly insightful.

F. Suggested improvements

The authors have comprehensively addressed all my previous suggestions:

- 1) Related approaches discussion: Now included in the Introduction with clear differentiation of FlexConsensus's unique positioning
- 2) Algorithm details: Fully specified with architecture, hyperparameters, and computational requirements
- 3) CryoBench validation: Excellent analysis of both IgG-RL and MDSpoke datasets with correlation to ground truth parameters
- 4) Resolution assessment: FSC curves demonstrate minimal impact of filtering on resolution
- 5) Figure quality: Improved to vector graphics with higher resolution
- 6) User accessibility: New Methods section on Scipion integration with tutorial links

G. References: appropriate credit to previous work?

Yes, appropriate references have been added.

H. Clarity and context

The manuscript is now significantly clearer with improved organization and the addition of practical workflow integration details.

Additional Comments

The authors provided thorough and thoughtful responses to my previous review comments. The additional experiments, particularly the CryoBench analyses, provided strong validation of the method. The correlation between consensus space and physical parameters (center-of-mass distance and dihedral angles for IgG-RL, collective variables for MDSpoke) convincingly demonstrates that FlexConsensus captures biologically meaningful information.

The practical improvements (Scipion integration, tutorials, interactive workflow visualization) will greatly enhance the tool's adoption by the cryo-EM community.

Remarks on code availability

The code is available through the provided GitHub repositories. The addition of tutorials and interactive workflow visualization significantly improves accessibility.